# Transcriptome analysis of archived tumors by Visium, GeoMx DSP, and Chromium reveals patient heterogeneity

Yixing Dong [1], Chiara Saglietti[2], Quentin Bayard [3], Almudena Espin Perez[3], Sabrina Carpentier[3], Daria Buszta [4], Stephanie Tissot [4,5], Rémy Dubois[3], Atanas Kamburov[3], Senbai Kang[1], Carla Haignere[3], Rita Sarkis [2], Sylvie Andre[4,5,6], Marina Alexandre Gaveta[4,5,6], Silvia Lopez Lastra[3], Nathalie Piazzon[2], Rita Santos[3], Katharina von Loga[3], Caroline Hoffmann[3], George Coukos [4,5,6], Solange Peters [4], Vassili Soumelis[3], Eric Yves Durand[3], Laurence de Leval [2,9] ✉, Raphael Gottardo [1,6,7,8,9] ✉, Krisztian Homicsko[4,5,6,9] ✉ & Elo Madissoon [3,9] ✉

Recent advancements in probe-based, full-transcriptome technologies for FFPE tissues, such as Visium CytAssist, Chromium Flex, and GeoMx DSP, enable analysis of archival samples, facilitating the generation of data from extensive cohorts. However, these methods can be labor-intensive and costly, requiring informed selection based on research objectives. We compare these methods on FFPE tumor samples in Breast, NSCLC and DLBCL showing 1) good-quality, highly reproducible data from all methods; 2) GeoMx data containing cell mixtures despite marker-based preselection; 3) Visium and Chromium outperform GeoMx in discovering tumor heterogeneity and potential drug targets. We recommend the use of Visium and Chromium for high-throughput and discovery projects, while the manually more challenging GeoMx platform with targeted regions remains valuable for specialized questions.

Recent advances in sequencing- and imaging-based techniques have led to the development of spatially resolved transcriptomics, named method of the year in 2020[1] with the ability to spatially quantify gene expression within a given tissue. These technologies allow health researchers and clinicians to characterize patient samples with unprecedented depth and spatial resolution, leading to transformative insights that have the potential to improve diagnoses, treatments, and patient outcomes.

Hybridization-based full-transcriptome methods relying on short probes have proven successful on FFPE tissue: Visium v2 (Visium)[2],

Chromium Flex (snRNAseq)[3] and GeoMx Digital Spatial Profiling (GeoMx DSP)[4] methods profile over 18,000 transcripts on the human genome, including the majority of the protein-coding transcripts necessary for the discovery of unconventional biological mechanisms and potential drug targets. Visium v2 enables uniform coverage of tissue with ~5000 of spots of 55 μm diameter, spaced 100 μm apart. GeoMx DSP allows users to pre-select regions of interest (ROI) on the tissue, and collect spatially distributed segments identified by fluorescence markers called areas of illumination (AOI). Chromium Flex is a single-cell/ single-nuclei transcriptomics platform with similar

[1]Biomedical Data Science Center, Lausanne University Hospital; University of Lausanne, Lausanne, Switzerland. [2]Institute of Pathology, Department of Laboratory Medicine and Pathology, Lausanne University Hospital and University of Lausanne, Lausanne, Switzerland. [3]Owkin, Paris, France. [4]Department of Oncology, Lausanne University Hospital; Swiss Cancer Center Leman, Lausanne, Switzerland. [5]Ludwig Insitute for Cancer Research, Lausanne, Switzerland. [6]Agora Translational Research Center, Lausanne, Switzerland. [7]Swiss Institute of Bioinformatics, Lausanne, Switzerland. [8]School of Life Sciences, Ecole Polytechnique Fédérale de, Lausanne, Switzerland. [9]These authors contributed equally: Laurence De Leval, Raphael Gottardo, Krisztian Homicsko, Elo Madissoon. ✉e-mail: Laurence.DeLeval@chuv.ch; Raphael.Gottardo@chuv.ch; Krisztian.Homicsko@chuv.ch; elo.madissoon@owkin.com

performance to the previous droplet-based methods, adapted for FFPE tissues[5]. Each individual method is both laboursome and costly to set up and perform. The relative strengths and weaknesses of the methods need better assessment to inform researchers on which technology to use.

While there have been significant recent advancements in spatial technologies (including Xenium[6] and CosMx[7]) enabling transcript quantification at cellular or even sub-cellular resolution, these technologies are often constrained by high costs, limited throughput, or restricted gene panels. Consequently, GeoMx DSP and Visium remain the most commonly used platforms for large-scale exploratory studies, such as MOSAIC[8]. Despite this, direct head-to-head comparisons of the two platforms remain scarce. There have been attempts at comparing Visium version 1 (v1) and GeoMx DSP. For instance, Wang et al. showed higher sensitivity but lower specificity on gene detection for GeoMx, using comparative experimental design for both methods[9]. However, since then, Visium was updated with an automated sample transfer tool, Cytassist, which offers improved sensitivity and specificity compared to its predecessor (Visium v1), making it the method of choice for the current study[10]. Furthermore, the release of single-cell RNA-seq for FFPE tissue on the Chromium platform has enabled its combined use with Visium to enhance the identification and mapping of malignant cell subtypes within single patients[3]. The synergistic application of Visium, GeoMx, and Chromium has been demonstrated in the creation of an atlas for cholestatic liver disease. In this work, Chromium was used to determine cell states, Visium to define tissue regions, and both Visium and GeoMx to confirm the co-localization of cell types within regions[11]. In a recent study, Yan et al. integrated GeoMx DSP, Visium spatial gene expression, and single-cell RNA sequencing (scRNA-seq) to comprehensively characterize tumor cell states and spatial cellular compositions within the tumor micro-environment (TME) of 19 patients, both before and after immunotherapy for non-small cell lung cancer (NSCLC)[12]. While this work highlights the complementarity of these approaches, it remains unclear which biological insights can be obtained from both spatial methods and which are exclusive to each technology.

Despite the successful application of Visium and GeoMx to cancer samples, a rigorous, large-scale head-to-head comparison of these technologies for high-throughput discovery projects across hundreds of patients is still lacking. Wang et al.[9], the only comparison study to date, includes just four breast cancer tissue samples and does not perform a direct comparison of Visium and GeoMx on registered adjacent tissue sections, as we do here. Furthermore, their study focuses on a non-specific set of immune AOIs (CD45 and CD8) and does not integrate snRNA-seq from adjacent sections with rigorous statistical methods.

In this work, we compare Visium and GeoMx in terms of operational performance, biological insights and propose optimal use-cases for each of these three technologies while taking advantage of the strength of each method. For this purpose we profile ROIs across four AOI labels (*Malignant*, *T cells*, *Macrophage*, *Other*) with GeoMx, as well as perform Visium and Chromium protocols on adjacent sections from 16 samples representing 14 patients with breast cancer, lung cancer, and DLBCL in archival FFPE blocks. By analyzing heterogeneity and intra-/inter-patient variations aided by tissue registration and deconvolution, we highlight the strengths, weaknesses and complementarities of the methods, as well as demonstrate how these insights can guide precision therapy.

## Results
### Experimental overview and data curation
First we generated a dataset of spatial and single-nuclei transcriptomes across three cancer types. Consecutive tissue sections of archival FFPE blocks (median age 57 months (22-103); median DV200 53.75% (7.2-80.3)) of breast and lung cancer resections (*n* = 4 each) and diffuse large B-cell lymphoma (DLBCL) excisional and sampling

biopsies (*n* = 6) were submitted for profiling with GeoMx, Visium v2 and Chromium FLEX platforms (Fig. 1a, Supplementary Fig. 1a, Supplementary Data 1-3). Tissue sections from each specimen with maximum size of 11mmx12mm were placed on GeoMx slides by groups of three and stained with fluorescence markers to collect the corresponding AOI-s (Fig. 1a, Supplementary Fig. 1a). For the Chromium FLEX workflow four samples were pooled in a single well. The nuclei were prepared using the snPATHO protocol[13] from two 25 uM cuts and were used for single-nuclei preparation. Wide distribution of different RNA quality values with DV200, histology types and block ages were chosen (Fig. 1b, Supplementary Fig. 1a). DV200 and the block age did not have a major impact on the gene detection, with even the sample B4 with considerably lower DV200 providing comparable data with other samples for Chromium and Visium (Supplementary Fig. 1b-e).

The number of data units varied across methods and samples (Supplementary Fig. 2a-e). GeoMx AOI-s were successfully selected for about 24 AOI-s per patient. Number of spots in Visium varied from 822 in a Breast cancer sample to 4951 in DLBCL depending on the size of the tissue. Chromium recovery was more variable from 802 nuclei to 17,804 nuclei per sample. We also observed high variation in the number of transcripts and in the number of genes detected across the samples with all three methods (Fig. 1c, Supplementary Fig. 1f).

To assess reproducibility, we compared data from adjacent sections in Breast (Fig. 1d). The samples performed consistently both in gene detection, as well as by clustering using high-dimensional representation. We observed a slide effect between replicates of lung samples in GeoMx, which was later corrected in downstream analysis (Methods). Overall, we conclude that the sample-level variability in data dominates the variability induced by processing, demonstrating high reproducibility for all three technologies. Overall, we present a highly reproducible spatial and single nuclei dataset characterizing the transcriptome of archival samples from three cancer types.

### GeoMx data exhibit non-specific signals in pre-selected AOI segments
We first aimed to compare GeoMx and Visium for their capacity to characterize cell types in tissue. For this purpose, we used image-based (i.e., H&E derived) labels for Visium spots and manually annotated Chromium data. AOI segment identity was used for GeoMx (Supplementary Fig. 2e, Supplementary Data 4a-c). For Visium, pathologists used the Loupe Browser (10X Genomics) to annotate all spots into tissue categories based on H&E images. These categories were then grouped into broader classifications: Tumor-enriched, Stroma-enriched, Lymphocyte-enriched, Immune Cell Mix (lung and breast), and Epithelia (DLBCL gastric and bronchial biopsies) (Supplementary Fig. 2c, Supplementary Data 5). The single-nuclei data was annotated into known cell types according to previously described markers and assigned to four hierarchical levels of cell type groups (Figs. 2a, Supplementary Fig. 3, 4, Supplementary Data 6, 7). Single-cell annotation at highest resolution (Level 4) enabled the identification of subtypes of malignant cells within one donor, effectively capturing intra-patient heterogeneity.

The cell-type composition of the AOI-s and spots was then derived by deconvolution of each data point using our Chromium reference data with the SpatialDecon[14] tool for GeoMx and Cell2location[15] for Visium (Methods). Both Visium and GeoMx methods captured enrichment of expected cell types in agreement to the Visium pathology annotation and the GeoMx curated AOI label (Fig. 2b, c). Also, in both methods tumor and stromal cell types were predominantly enriched in their respective Tumor and Stromal regions. T-cell signatures were enriched in *T cells* AOI-s in GeoMx, and the T- and B-cell signatures were enriched in Lymphocyte regions in Visium.

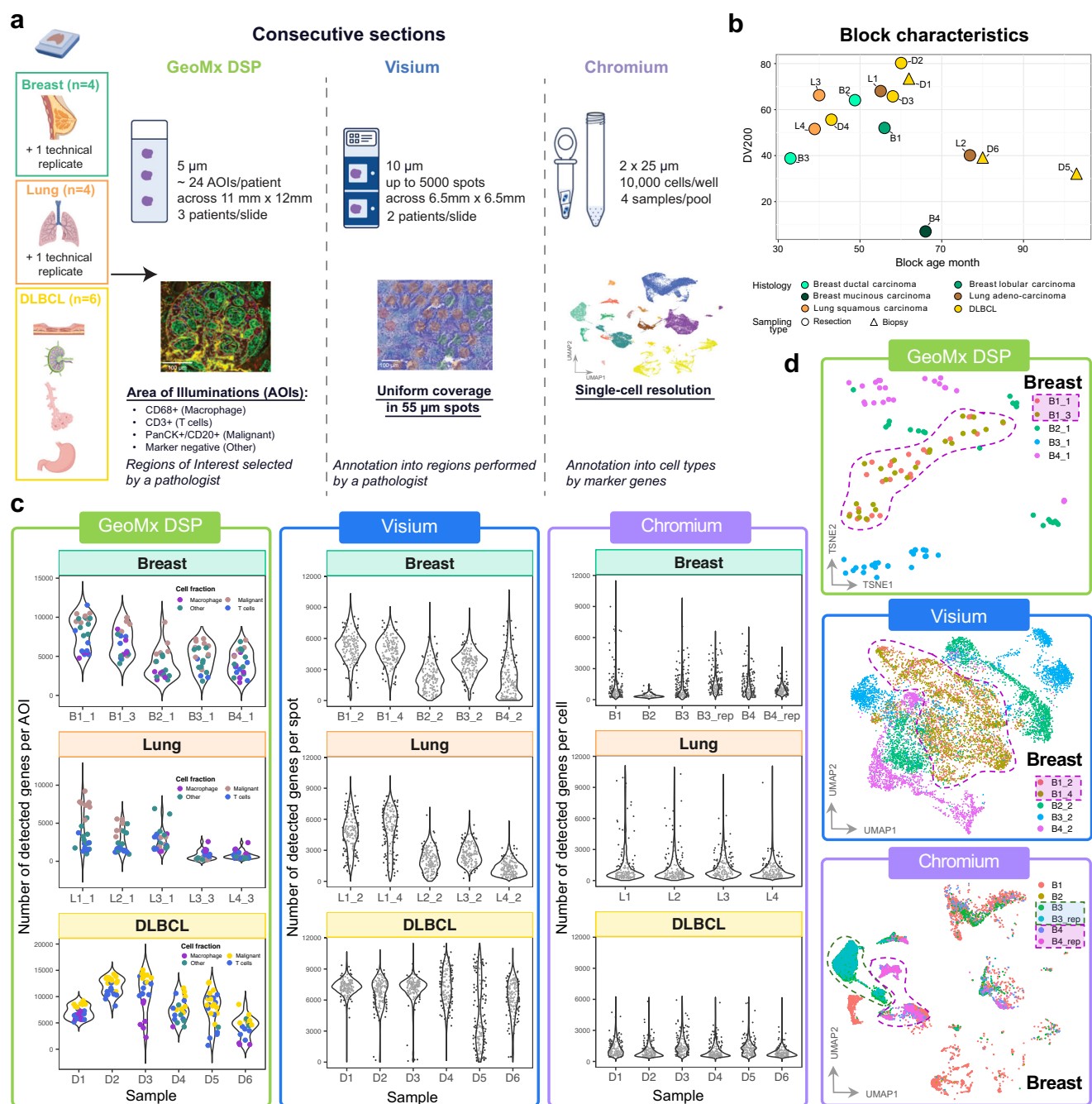

**Fig. 1 | Experimental setup and data quality with three full transcriptome methods GeoMx, Visium and Chromium on Breast and Lung cancer, and DLBCL biobank samples. a** Schematics of experimental design to generate GeoMx and Visium spatial transcriptomics and Chromium single-nuclei data for three cancer types corresponding to 14 donors from FFPE blocks with 2 samples replicated for each technology. AOI - area of illumination, DLBCL - Diffuse large B-cell lymphoma, DSP - digital spatial profiler. Created in BioRender. Gottardo, R. (2025) https://BioRender.com/o13c311. **b** Display of block age and RNA quality measure DV200 values for all the samples with histology type, labelled by patient ID. **c** Gene detection rate for GeoMx AOI-s and number of genes detected in Visium spots and single nuclei. Colors in GeoMx correspond to AOI labels. **d** Breast cancer tSNE plots with GeoMx and UMAP plots with Visium and Chromium show good reproducibility. Technical replicates with adjacent sections from the same block are highlighted with a dashed line. GeoMx: 5 samples/117 AOIs; Visium: 5 samples/10249 spots; Chromium: 6 samples/10689 cells. B breast, L lung, D DLBCL. Source data are provided as a Source Data file.

*Macrophage* AOI-s in GeoMx and Immune cell mix in Visium also matched expected cell type enrichments. In addition to the expected signal, there was an unexpected stromal signal for all the immune-related regions in both methods. This was expected in Visium where, by design, spots capture around 20 cells. In GeoMx, nonspecific signal in AOI capture was also observed with canonical markers' expression in Breast, Lung, and DLBCL (Supplementary Fig. 5a, b). The cell type specificity on a global scale was lost in the Breast *T cells* that clustered together with *Other* segments (Supplementary Fig. 5c). Although the GeoMx method was designed to capture single cell types, we observed that the least specific signals originated from AOIs containing scattered cells, such as *T cells*. In contrast, the purest signals were obtained from tissue areas with high uniformity and a large surface area relative to the AOI contour (Fig. 2d). The nonspecific signal from Visium is explained by the spots laying at intersection between two areas, and cell type mixtures positioned within the same area (Fig. 2e).

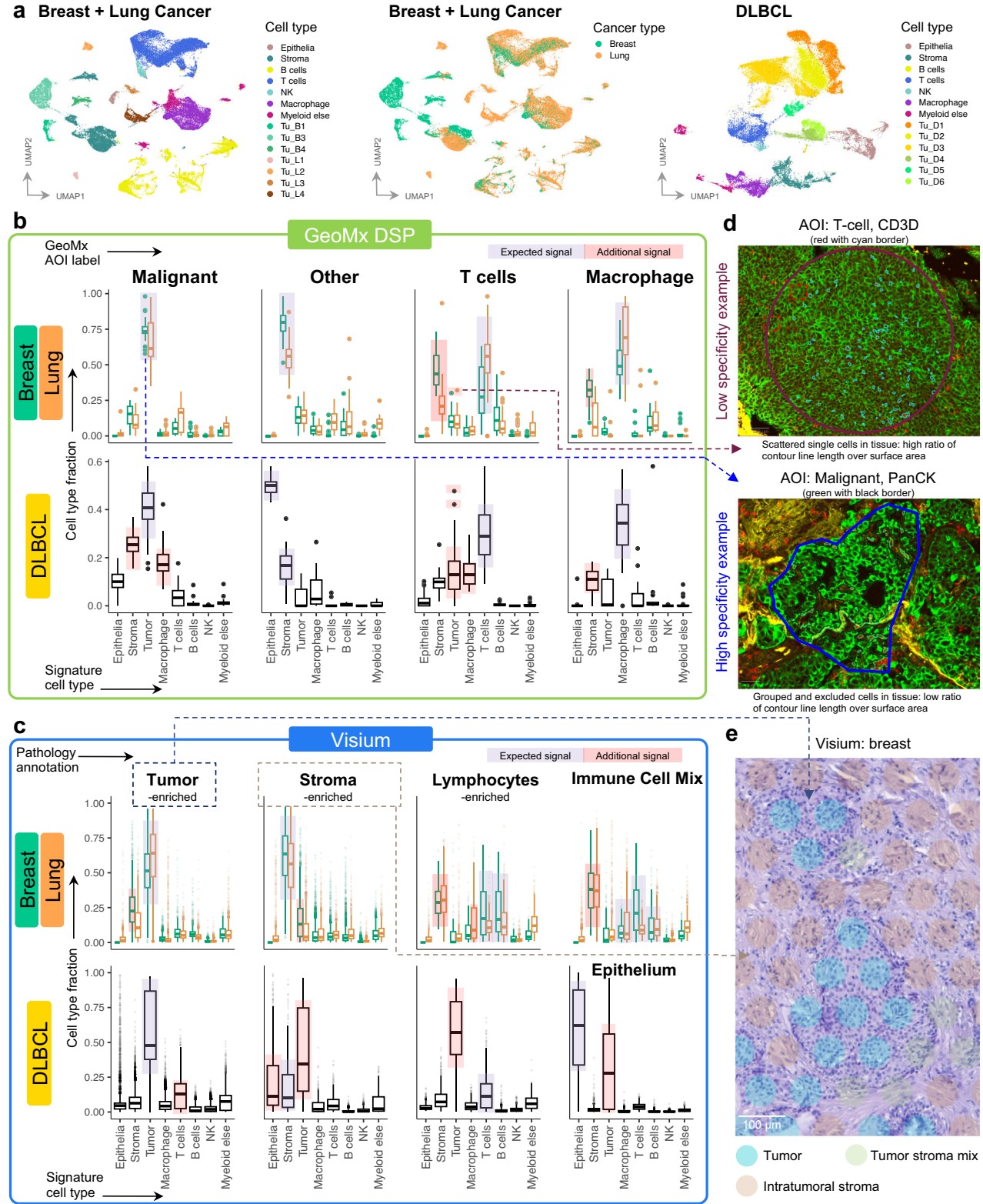

Additionally, our H&E-derived labels assigned spots based on their predominant cell type(s), so background signals from other cell types were expected (Methods). Overall, we concluded that both spatial methods captured mixtures of cells that can be readily deconvolved computationally. However, in the case of GeoMx, the selection of AOIs within cell clusters versus scattered cells in the tissue can influence the specificity of the signal.

## Cell type specificity in matched GeoMx and Visium regions

To directly compare GeoMx and Visium, we aligned images to match Visium spots with specific AOI locations in GeoMx (Methods, Fig. 3a). In total, nine pairs of consecutive sections of GeoMx and Visium breast and lung samples were used for alignment (Supplementary Fig. 6a). When 70% of the Visium spot's area was overlapping with a GeoMx segment, it was considered matching to that segment's AOI label.

**Fig. 2 | Assessment of cell type specificity in GeoMx AOI-s and Visium spots.**
**a** UMAP plots for annotated Chromium data at Level 1.5 resolution. Breast & Lung: 10 samples/46,643 cells; DLBCL: 6 samples/39,713 cells. Predicted cell type fractions by deconvolution in GeoMx (**b**) and Visium (**c**), grouped by AOI label in GeoMx and Pathologist annotation groups in Visium. Fractions are displayed for groups of shown cell types. Boxes represent the quartiles of the data, while the whiskers extend to data points within 1.5x the interquartile range from the lower and upper quartiles. The horizontal black line within each box represents the median. Outliers are indicated as individual data points. GeoMx: Breast (*Malignant*: n = 225 AOIs; *Other*: n = 281 AOIs; *T cells*: n = 148 AOIs; *Macrophage*: n = 134 AOIs), Lung (*Malignant*: n = 248 AOIs; *Other*: n = 296 AOIs; *T cells*: n = 296 AOIs; *Macrophage*: n = 104 AOIs), DLBCL (*Malignant*: n = 448 AOIs; *Other*: n = 72 AOIs; *T cells*: n = 416 AOIs; *Macrophage*: n = 152 AOIs). Visium: Breast (Tumor-enriched: n = 15,352 spots;

Stroma-enriched: n = 36,086 spots; Lymphocytes-enriched: n = 638 spots; Immune Cell Mix: n = 1625 spots), Lung (Tumor-enriched: n = 15,352 spots; Stroma-enriched: n = 13,224 spots; Lymphocytes-enriched: n = 7160 spots; Immune Cell Mix: n = 14,088 spots), DLBCL (Tumor-enriched: n = 120,824 spots; Stroma-enriched: n = 5288 spots; Lymphocytes-enriched: n = 5840 spots; Epithelium: n = 7592 spots). Expected and unexpected additional signals for that AOI label or pathology group are highlighted. Pathology annotations were grouped as in Supplementary Data 5. **d** Examples of AOI-s with low and high specificity on immunofluorescent images. Thicker line is contour for the full region of interest, thinner line is contours for AOI. Images above are representative of 1 sample. **e** Examples of Visium spot annotation into Tumor and adjacent areas. Image above is representative of 1 sample. Source data are provided as a Source Data file.

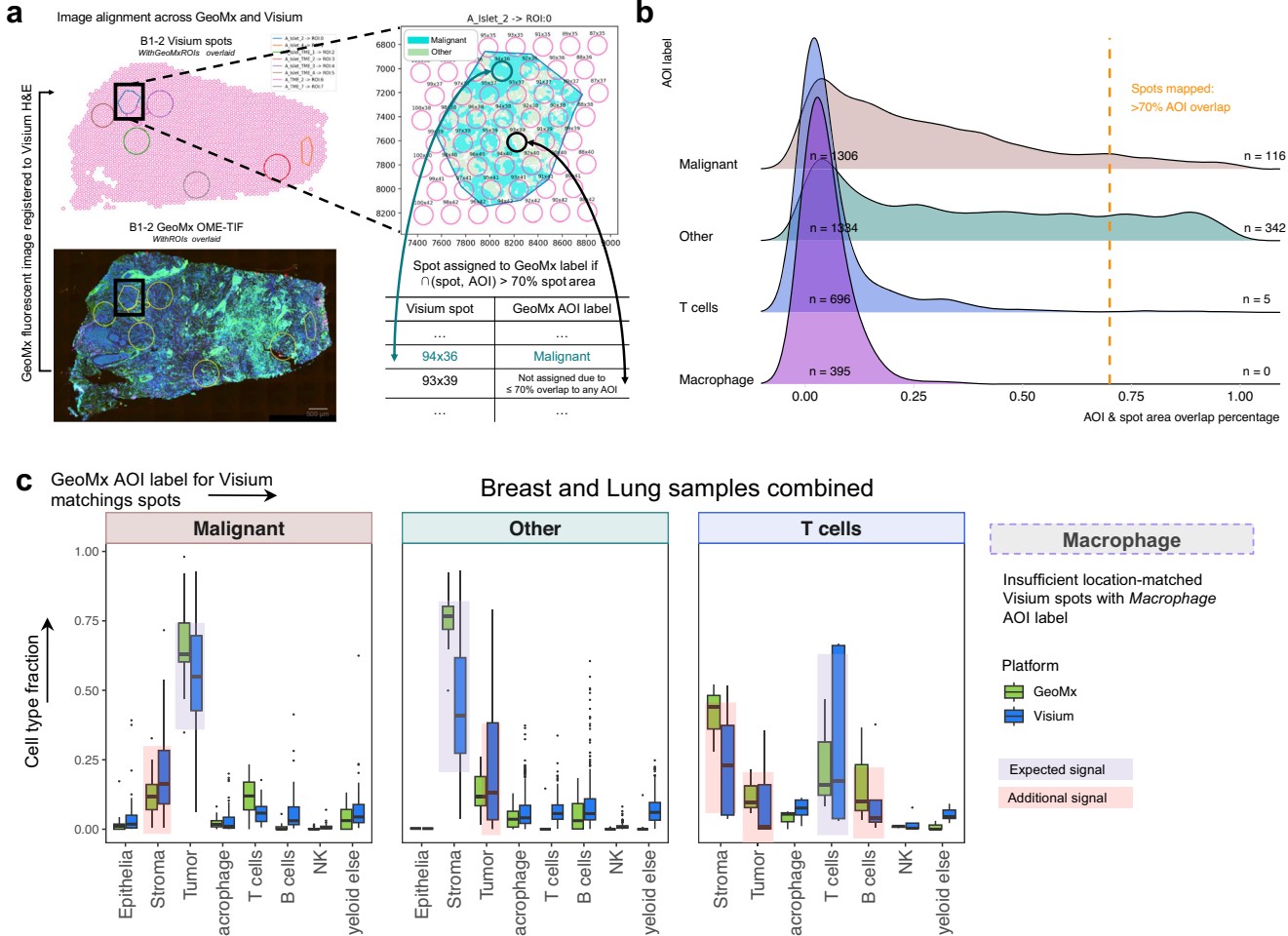

**Fig. 3 | Cell-type deconvolution specificity comparison on spatially registered Visium spots to GeoMx AOI labels.** **a** Image registration between GeoMx fluorescent image and Visium spot coordinates. Image above is representative of 1 sample. **b** Density plot of overlapping area between Visium spots and each GeoMx AOI. A threshold of >70% was applied to select spots registered to a GeoMx AOI label. Number of mapped spots: n total (n passed 70% threshold)−*Malignant*: n = 1422 (116 AOIs); *Other*: n = 1676 (342 AOIs); *T cells*: n = 701 (5 AOIs); *Macrophage*: n = 395 (0 AOIs). (**c**) For registered GeoMx segments and Visium spots in (**b**), using

matched Level 1.5 Chromium reference from Fig. 2a to deconvolute cell type fraction. Boxes represent the quartiles of the data, while the whiskers extend to data points within 1.5x the interquartile range from the lower and upper quartiles. The horizontal black line within each box represents the median. Outliers are indicated as individual data points. *Malignant*: GeoMx (n = 28 AOIs), Visium (n = 116 spots); *Other*: GeoMx (n = 29 AOIs), Visium (n = 342 spots); *T cells*: GeoMx (n = 3 AOIs), Visium (n = 5 spots). Source data are provided as a Source Data file.

Around 11% of the spots that fell into an AOI got spatially mapped to an AOI label. *Malignant* and *Other* AOI-s had the highest number of spots mapping to their area providing 116 and 342 data points respectively (Fig. 3b). Spots mapping to either *T cells* or *Macrophage* AOI-s had a smaller overlapping area to a GeoMx segment, providing only five spots for *T cells* and no matching spot for *Macrophage* AOI label. This is

due to the gaps between Visium spots, and the irregular shapes of immune cell marker staining in GeoMx that fail to be captured by Visium spots. For the location-matched spots, we compared the cell type specific signal with deconvolution between GeoMx and Visium (Fig. 3c). Both technologies showed comparable cell type fraction in *Malignant* and *Other* AOI-s. In matched *T cells* AOI-s, we saw a higher

proportion of stroma cells than T cells in GeoMx, while such a background signal is suppressed in Visium. Note that all 5 mapped spots belonged to breast samples (Supplementary Fig. 6d), for which the stromal signal is lower than T-cell signal compared to that in lung samples (Fig. 2b). To estimate the signal from these minority cell types in GeoMx and Visium independently of location matching, we display the T-cell and macrophage deconvolution fractions of each AOI or spot on integrated reduced dimension plots for all indications (Supplementary Fig. 7c, d). Lung samples show higher abundance of T-cell and macrophage spots in Visium, consistent with cell type proportions in Chromium (Supplementary Fig. 4). However, we observe a higher deconvolution fraction of T-cell and macrophage in GeoMx' manually selected AOI-s (Supplementary Fig. 7d). This demonstrates the strength of highly segmented GeoMx design that allows a better enrichment for rare cell types of interest in a tissue.

Overall, we conclude that Visium and GeoMx demonstrate comparable specificity in head-to-head spatial registration comparisons and effectively capture signals from rare cell types. Notably, despite the enhanced cell-type enrichment offered by GeoMx's AOIs, Visium reliably detects signals from corresponding regions.

## Deconvolution into cell types enhances spatial characterization in Visium and GeoMx

Next, we explored data-driven cell type composition through deconvolution in both Visium and GeoMx, and compared them to the H&E-driven pathology annotations or morphology marker-based segments correspondingly.

The deconvolution of each Visium spot into cell type fractions enables higher resolution of the tissue than distinguished by pathologists based on H&E alone (Fig. 4a, b). Cell type fractions derived from the deconvolution of GeoMx AOIs vary across regions and also exhibit significant variability within a single segment type (Fig. 4c, d). Inspections of the AOI pie charts reveals that the variation for selected cell types ranged from about 5% to about 35% of the predicted T-cell abundance in the *T cells* AOI and from less than 25% to over 45% of the predicted macrophage abundance in the *Macrophage* AOI. These findings highlight the presence of mixed cell populations, despite the segments being primarily defined for a single cell type in donor B1.

The pathology label and deconvolution majority vote for all Visium samples were shown in a gallery (Supplementary Figs. 6b, c, 7a, b). Across all Breast, Lung and DLBCL donors, both Visium and GeoMx demonstrated overall consistency in the tumor cell capture compared to the pathology labels or segment types correspondingly (Fig. 4e, f). T- and B-cell abundances were highest in the annotated lymphocyte-enriched or Immune Cell Mix regions in Visium as expected, as well as the stromal cell abundances in the stromal regions. T-cell and Macrophage cell type abundances were more specific in the *T cells* and *Macrophage* AOIs in GeoMx for all three indications. High stromal cell abundance was observed for both methods in all the annotated or segmented regions. A further breakdown of such agreement analysis with Chromium Level 4 annotations were shown for breast, lung, and DLBCL (Supplementary Fig. 8a, b).

We then focused on regions where Visium deconvolution showed increased resolution compared to H&E-based pathology annotations, and used GeoMx as a validation by spatially registering the matching sections. The increased resolution by deconvolution is evident in regions with similar histology but different cell type composition. First, T- and B- cells were annotated as Lymphocytes on H&E, but deconvolution can distinguish these cell types in Visium (Fig. 4g). In the corresponding GeoMx section, the B-cell signal is abundant across the *Other*, *T cells*, and *Macrophage* AOI-s, despite not specifically staining for *"B cells"* AOI. B cells notably co-exist with macrophages, as observed in both Visium spot deconvolution and GeoMx AOI pie charts. For the T-cell region identified in Visium, no GeoMx ROI was sampled, and thus the gene expression data is not available for GeoMx. Second, malignant cell

subtypes identified in Chromium for patient D3 showed spatially distinct locations between Tu_D3_FAM3C and Tu_D3_dividing on Visium (Fig. 4h). GeoMx showed consistent localisation in the Tu_D3_dividing region, but was missing an ROI in the Tu_FAM3C region. Finally, neoplastic B-cell and stomach epithelium layers in DLBCL biopsy are clearly separated by deconvolution into distinct regions, although intermixed with pathology annotations (Fig. 4i). Reasonable spatial concordance is observed between Visium and GeoMx in the four selected ROI-s for the tumor and epithelium subtypes.

Overall, we demonstrate that data-driven annotation via deconvolution offers significantly higher resolution compared to threshold-based segmentation in GeoMx and manual H&E-based annotation for Visium. While we observe reasonable concordance between the two technologies, the manual and subjective selection of a limited number of ROIs and their associated AOIs poses substantial challenges for analysis and visualization. This selective process restricts the dynamic range of gene expression for certain targets, making quantification more difficult. In contrast, Visium samples the entire tissue, capturing spots with both low and high expression levels, thereby enabling more meaningful and comprehensive quantification.

## Spot-level data can be enhanced computationally to increase resolution and detection rates

A common criticism of the Visium technology is its resolution and data sparsity[1]. Each spot typically contains an average of about 20 cells, and similar to single-nuclei RNA-seq, the expression levels for canonical marker genes can be relatively low, making it challenging to identify and visualize detailed tissue structures. To address this limitation, we applied BayesSpace[16], a reference-free approach that subdivides each spot into subspots, estimating the gene expression contribution of each subspot to the overall spot-level value, thereby generating a super-resolution image. With its high-resolution gene expression maps, BayesSpace can resolve tissue structures that are difficult to detect at the original resolution. To illustrate this, we analyzed a sample containing a visible tertiary lymphoid structure (TLS) on the H&E image. The corresponding area in Visium shows limited evidence of TLS, with sparse expression of canonical marker genes such as *CXCL13*, *MS4A1*, and *CD4* (Fig. 5a). However, after enhancement with BayesSpace, the same region reveals a well-defined area with elevated expression of these marker genes. These findings show that the resolution of Visium data can be significantly enhanced, thereby increasing its analytical value for spatial profiling.

To determine whether a similar biological signal exists in the consecutive GeoMx section, we analyzed the ROI corresponding to the same tissue location where the TLS was identified in Visium. We expected TLS markers to be highly expressed in the "Other" segment of the ROI in TME compared to neighboring malignant ROIs. However, with the exception of *CXCL13*, most markers did not show significant expression. As discussed earlier, unlike Visium, which provides full spatial coverage with uniform spot sizes, GeoMx AOIs are more limited in number and can vary significantly in size, while averaging the expression across the whole capture area. Consequently, the deconvolution fraction serves as a more reliable indicator, offering internal normalization across all AOIs and providing a cleaner signal. Using this approach, we observed a higher B-cell deconvolution fraction in the ROI overlapping with TLS compared to other ROIs (Supplementary Fig. 9a).

In addition to the previously mentioned TLS markers, we analyzed an additional sample, L4, to compare the sensitivity of Visium and GeoMx in detecting immune cell signals. We focused on an immune cell aggregate identified through pathology annotation on Visium, matched to an ROI in GeoMx (Supplementary Fig. 9b). At the spot-level resolution in Visium, only *CD14* exhibited a clear signal. However, using BayesSpace-enhanced subspot resolution, numerous canonical markers for B-cells, T-cells, and macrophages highlighted the immune cell aggregate. In contrast, the immune cell signals in GeoMx were not

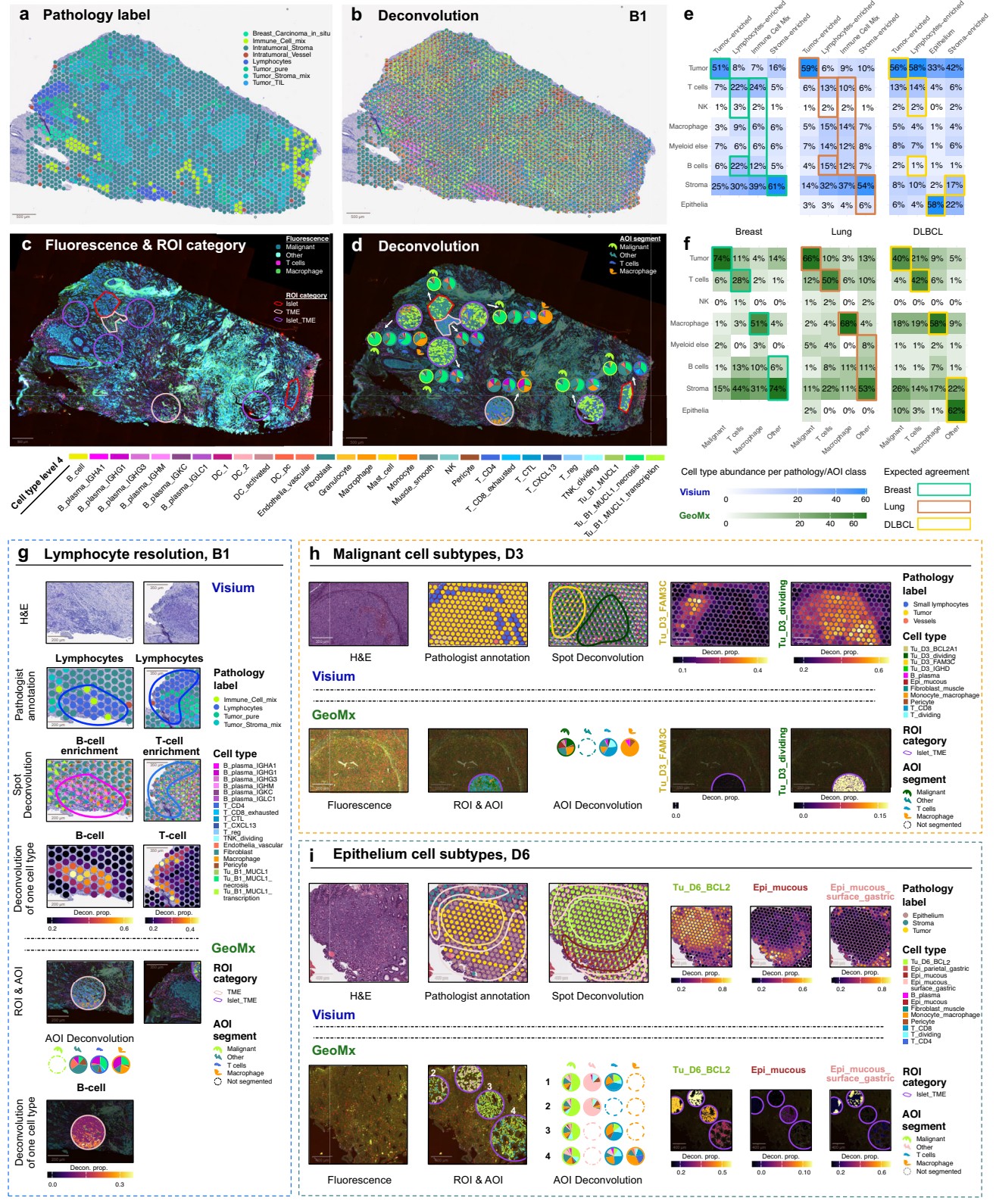

consistently stronger in the TME compared to the Islet in the expected segment. Deconvolution fractions for B cells, T cells, and macrophages in GeoMx showed expected intensities based on their AOI locations but lacked the ability to pinpoint specific regions due to the pre-defined irregular shape of the ROI-s (Supplementary Fig. 9c). Overall we demonstrate that Visium has the higher capacity to detect structures that were not pre-defined before study, allowing for a better hypothesis-free approach to analysis.

## Visium and GeoMx reveal spatially distinct malignant regions with unique transcriptional profiles

In addition to identifying structures, we aimed to look for intra-patient variation in Visium and GeoMx. Using the spatially-aware clustering method, BayesSpace, we discovered two transcriptionally distinct areas on the reduced dimension UMAP for patient B3 on Visium (Fig. 5b). The regions had similar morphology, annotated as "Tumor_pure" and received a cell type majority vote "Tu_B3". We group

**Fig. 4 | Multi-modality of Visium and GeoMx data visualized with H&E and fluorescence image, pathology label, and deconvolution. a** Pathology label visualized on H&E image of Visium sample B1_4. **b** Deconvolution cell type fractions plotted as scattered pie charts for each spot visualized on H&E image of Visium sample B1_4. **c** Pathological regions of interest (ROI-s) visualized on fluorescence images of GeoMx sample B1_3. **d** Up to four AOI segments have their outlines complement each other in an ROI. Deconvolution cell type fractions plotted as pie charts for each area of illumination (AOI) segment within each ROI. Sample B1 - Visium: 1988 spots; GeoMx: 24 AOIs. Pathology label and deconvolution agreement for breast, lung and DLBCL in Visium (**e**) and GeoMx (**f**) samples. Heatmap is labelled by the average cell type deconvolution fraction per pathology label or AOI label, colored by square root of the number. **g** Lymphocyte resolution zoomed in on H&E image, pathology annotation, deconvolution shown as scatterpie of cell type fractions, and individual deconvolution fraction for cell type of interest in

Visium sample B1_4. At the same spatial location, ROI and AOI outlines (if any) were shown on the consecutive section of GeoMx sample B1_3. The individual deconvolution fraction of B-cells is shown spatially in the TME ROI. The T-cell region identified in Visium did not have an ROI sampled exactly head-to-head in GeoMx. **h** Malignant cell subtypes visualized across various modalities for consecutive sections of Visium and GeoMx sample D3. Deconvolution is able to depict the spatial transitioning in Visium of tumor cell types annotated in Chromium. The absence of Tu_D3_FAM3C and the existence of Tu_D3_dividing can be validated in GeoMx at the same spatial location. Sample D3 - Visium: 4951 spots; GeoMx: 23 AOIs. **i** Tumor and epithelium cell subtypes visualized across various modalities for consecutive sections of Visium and GeoMx sample D6. Sample D6 - Visium: 1649 spots; GeoMx: 19 AOIs. Images above are representative of 3 samples. Source data are provided as a Source Data file.

---

the clusters as cluster 14 (Area A) and clusters 1, 5, and 9 (Area B). For all AOIs in GeoMx sample B3, we mapped their spatial location to Area A and B. In concordance to Visium, the malignant AOIs also showed separate clusters in the UMAP of GeoMx, by Area A and B. The deconvolution result indicates that more than 60% of the *Malignant* AOI-s are composed of Tu_B3 cell type (Fig. 5c). We then performed Differential Expression (DE) analysis between the two tumor areas in Visium and GeoMx (Methods). Not surprisingly, due to the larger number of spots and increased statistical power, Visium identifies more significantly differentially expressed (DE) genes and higher fold-changes compared to GeoMx. The majority of the top DE genes identified by GeoMx are also detected by Visium (Fig. 5d, e). Some of the DE genes identified are drug targets (Methods), for example, a potential treatment for marker *GSTP1* in Area B is ezatiostat and for marker *ADRA2A* in Area A is dexmedetomidine (Fig. 5d). This intra-patient heterogeneity was replicated with Chromium data where we annotated two tumor subclusters for Tu_B3 corresponding to the area A and B with specific DE genes *PLA2G2A* and *NPPC* correspondingly (Supplementary Fig. 9d). The two subclusters mapped to the expected areas in Visium, and the DE analysis on Chromium showed similar significant DE genes as the spatial analysis (Supplementary Fig. 9e, f). We next examined the consensus of differentially expressed (DE) genes across the three platforms in each area. Visium identified the highest number of distinct DE genes, while GeoMx identified the fewest (Fig. 5h). Notably, greater overlap was observed between Visium and GeoMx, as well as between Visium and Chromium, highlighting the strong potential of integrating Visium with other platforms for uncovering intra-patient heterogeneity.

## Potential for targeted therapies in heterogeneous patient population

Besides molecular characterisation of intra-patient heterogeneity in Breast, we wanted to explore the power of full transcriptome methods to describe existing drug targets and discover potential ones in a highly heterogeneous and complex tissue such as DLBCL. We first aggregated annotations of DLBCL malignant and tumor microenvironment (TME) cells to similar categories (Fig. 6a) by using Level 2 categories in Chromium with malignant subtypes, annotating Visium spots and using GeoMx AOI labels. We did observe patient-effect in the uncorrected data for both Visium (Supplementary Figs. 10, 11) and GeoMx (Supplementary Fig. 12 b, h, n) also in the non-malignant spots/ AOI labels. After adjusting for donor effect (Methods), Visium annotation into epithelium, stroma, plasma, and vessels/immune was done using clustering and known marker gene enrichment (Methods, Supplementary Fig. 13). We used canonical marker expression and relied on pathologist annotations for Necrosis regions. Malignant nuclei were annotated by both enrichment of specific patients in the cluster as well as the cell type signature similarity with the corresponding donors' malignant cell markers gained from Chromium data (Supplementary Fig. 4b–d).

Known and potential drug targets were selected manually from known targets or top targets for exploring intra- and interpatient variability (Fig. 6b, Methods). In fact, many of them are among the top 100 most differentially expressed genes when comparing malignant cells or spots to their non-malignant counterparts (Supplementary Fig. 14, Methods). The B-cell markers were enriched in all patients' malignant cells compared to the TME cells with Chromium data. We observed differences in expression between patients for many drug targets including *CD47*, *CD52*, *CD40* and *CD38*, potentially suggesting an existing patient subgroup that might better respond to antibody-drug conjugates (ADC) designed against those surface molecules. Interestingly, we also observed intra-patient variability in expression in some potential drug targets. *CD52* and some *BCL* genes showed varying expression within patient D3 between the identified subclones in single-nuclei data. Other potential drug targets such as *FCRL* genes showed patient-specific expression in D1, D2, D3 and D6. Interestingly, patient D5 stood out with very specific gene expression patterns, suggesting more drastic differences in the underlying biology.

While there was considerable heterogeneity both within and between patients' malignant nuclei in the Chromium data, these differences are more difficult to detect in the Visium or GeoMx methods, likely due to the mixing of cells within spots or AOI-s. We tried to enhance the cell type signal specificity in Visium and GeoMx by using deconvolution information with Chromium. Instead of the clustering approach above, we used majority votes from deconvolution, and showed densities of each cell type fraction for its corresponding majority vote class (Supplementary Fig. 15a, b). To enhance the purity of spots, only spots with more than 50% maximum cell type fraction were kept in Visium (Supplementary Fig. 15b). After such filtering, 52% DLBCL spots were included in the downstream analysis for Visium. For GeoMx, we kept 75% segments that showed consensus between AOI label and deconvolution majority vote label (Supplementary Fig. 15c, d). The "purified" Visium and GeoMx data revealed a more distinct pattern in the expression levels of known drug targets, more similar to the patterns observed in Chromium, particularly for patient D5 (Supplementary Fig. 15e). However, a lack of cell type specific gene expression persists for GeoMx.

This use case is prominent in DLBCL where cellular density in tissue is high and heterogeneous, demonstrated by higher additional signals compared to breast and lung (Fig. 2a, b). For tissues with mixed cell types, Chromium and Visium offer the highest resolution, with Chromium providing superior granularity. This underscores the critical importance of integrating matched single-cell or single-nuclei RNA-seq data with full-transcriptome spatial data to comprehensively capture the complexity of cellular heterogeneity.

## Discussion

We aimed to compare full transcriptome methods on archival FFPE blocks from tumor patient samples: GeoMx DSP, Visium CytAssist and Chromium Flex. We assessed the operational challenges, specificity of

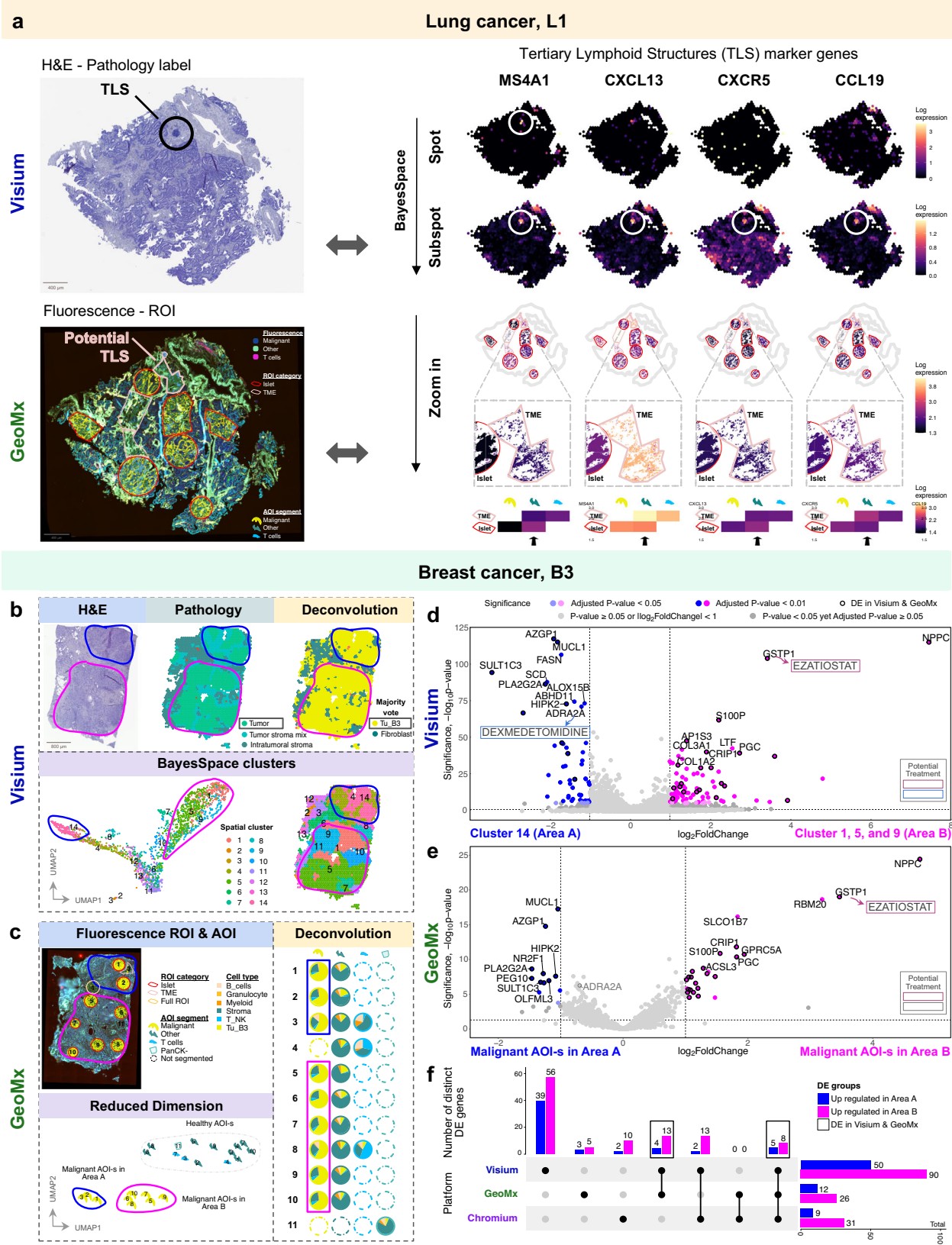

**a** Lung cancer, L1

H&E - Pathology label

Tertiary Lymphoid Structures (TLS) marker genes

MS4A1    CXCL13    CXCR5    CCL19

**b** Breast cancer, B3

cell type signatures, ability to capture heterogeneity and potential for high-throughput set-up.

The operational setup of GeoMx required more resources in terms of optimization, cost, and lab personnel training compared to Visium and Chromium. However, it offered greater flexibility in experimental design, despite being more susceptible to batch effects

(Table 1, Supplementary Fig. 12). Chromium had about twice lower running cost, making the user consider whether spatial resolution on cell type groups is twice more valuable than corresponding single-cell/nuclei resolution. However, Chromium is suggested to consume at least five times more tissue than the spatial methods[3] (Table 1), making this an important consideration for low-input clinically valuable

**Fig. 5 | Integrating H&E image, pathology label, deconvolution, and clustering identifies TLS and tumor subtypes in Visium. a** For lung sample L1, TLS markers show boundary of TLS after BayesSpace resolution enhancement, which validates the TLS location by pathology label in Visium. In the consecutive section of L1 in GeoMx, at the same tissue location where the TLS was identified in Visium, we zoomed in to check the TLS markers expression level in the TME ROI compared to its neighboring Islet ROI. Black arrows point to the *Other* AOI, where the B-cell signal is expected to appear. Sample L1 - Visium: 944 spots; GeoMx: 25 AOIs. **b** For breast sample B3, two regions were similar in H&E, labeled as tumor by pathologist, and deconvolution majority voted as single tumor type from Chromium; two spatially distinct tumor subtypes appeared with spatially-aware clustering in Visium. **c** In the consecutive section of B3 in GeoMx, the ROI-s were mapped to the two regions identified by Visium. The *Malignant* AOI-s show distinct clusters in the reduced dimension. Deconvolution results for each AOI are displayed as pie charts. Sample B3 - Visium: 2156 spots; GeoMx: 23 AOIs. **d** Volcano plot of Differential Expression (DE) analysis between selected clusters from (**b**) (moderated *t*-test, two-sided). Genes for which there are existing drugs are highlighted (see further information in Supplementary Data 8). **e** Volcano plot of DE analysis between two *Malignant* AOI clusters in (**c**) (Wilcoxon's Rank Sum test, two-sided). Genes that are mutually DE in both Visium and GeoMx are highlighted with black outline. **f** Upset plot showing number of distinct DE genes identified by Visium, GeoMx, and Chromium, and their combinations. Images above are representative of 2 samples. Source data are provided as a Source Data file.

samples such as core-needle biopsies. Estimated maximum throughput per week is highest for Chromium and lowest for GeoMx with the current experimental set-up. Pre-processing and data analysis can rely on vendor-provided tools, but advanced analyses are easily available for Visium and Chromium with a large scientific community and contributors but require more specialized effort for GeoMx.

Although the data was highly reproducible across similar biological samples for all three methods, GeoMx and Visium exhibited greater biological variability across samples due to the mixing of cells within individual data points. This variability is inherent to Visium, as its capture of transcripts within circular spots on the tissue naturally leads to cell mixtures, which was expected. However, it was surprising to observe high fractions of other cell types in the GeoMx AOI-s, which were preselected for specific cell types—defeating the purpose of targeted selection to some extent. This issue was particularly pronounced in scattered cell types within the tissue, potentially due to transcript or probe leakage from nearby areas or physically overlapping cells along the *Z*-axis of the tissue. Contamination in GeoMx could be reduced with a simpler experimental design, such as selecting fewer segments or better-separated cell clusters, but making the flexibility of GeoMx a less relevant advantage of the method.

While the challenges of setup and unexpected transcript signals in GeoMx could have been mitigated with a simpler design, we aimed for the highest resolution of cell types, as this was the biggest advantage of GeoMx over Visium. We collected CD68-enriched data from GeoMx that were not matched with Visium, highlighting the benefit to address specific questions. The current best full-transcriptome resolution would be the Visium HD method with a $2 \times 2 \, \mu m$ grid[17]. However, this method is much more expensive, making it less applicable to larger studies. The highest spatial resolution would be provided by in situ sub-cellular CosMx and Xenium methods that can detect up to 6000 genes[6,7] and more recently even full-transcriptome[18]. However, the costs for these in-situ technologies are even higher, and their throughput is at multiple days per sample, not allowing scale-up for study across a large number of patients. The development of subcellular technologies into full-transcriptome assays is the next frontier in studying tissue biology, with the cost and analysis difficulty hindering the usage by scientific and clinical communities.

We also demonstrated the discovery of transcriptional heterogeneity on morphologically similar tissue with both Visium and GeoMx. This finding underscores the unique potential of Visium, Chromium, and GeoMx to independently uncover intra-patient heterogeneity that may be overlooked by standard pathology relying on H&E image or marker-based annotation. This enabled identification of two regions expressing targets for two different drugs, thereby demonstrating the potential for combination therapies for specific patients. Notably, both Visium and GeoMx provide the added capability to spatially map these subtypes, offering deeper insights into tissue architecture and cellular distribution. While there is reasonable agreement in identifying top targets across platforms, Visium's higher resolution and larger sample size enabled improved detection of spatially distinct regions and their associated transcriptional profiles,

thanks to its uniform and unbiased sampling approach. One approach to reduce sampling bias in GeoMx is to increase the number of selected ROIs. However, this comes with a trade-off, as the cost of GeoMx analysis scales with the number of data points. Additionally, while an AOI typically captures higher read counts averaging transcripts across a wider tissue area than a Visium spot, the signal within the whole AOI region is uniform, not enabling to pinpoint the location of specific transcriptome or tissue region. This limitation is particularly evident in zoomed-in regions, where AOI-derived fractions are consistently less sensitive compared to Visium spots.

Finally, we explored the functionality of full transcriptome methods in drug target discovery from highly heterogeneous FFPE tissue on DLBCL. Expression of known and potential drug targets varied across patients and within patients with Chromium data, showing promise for patient subgroup identification and for assigning personalized and combination therapies. Expression differences were less distinct in Visium and even less so in GeoMx, likely due to the presence of mixed cell types and the larger capture areas associated with GeoMx. Additionally we observed substantial donor-to-donor variability, further emphasizing the utility of these platforms for identifying targeted therapies tailored to individual donors. While all three methods have the potential for the discovery of targets, GeoMx is limited in the number of data points for intra-patient heterogeneity discovery, and both GeoMx and Visium are limited in cell type specific signal.

Overall, we successfully applied full-transcriptome methods in tumor FFPE blocks and demonstrated the applicability of Visium and Chromium for the systematic spatial profiling of large panels of samples. The unbiased nature of the two platforms make them ideal in high throughput setups aiming at characterizing thousands of tumor samples. Based on this data, we propose Visium and Chromium as the platforms of choice for spatial and transcriptomic analysis of tumor samples as part of the MOSAIC (Multi-Omics Spatial Atlas in Cancer) project[7]. The MOSAIC consortium will systematically profile thousands of tumor samples to analyze their spatial and single cell structures. By integrating these data with other modalities, like H&E staining, using advanced AI, we aim to create meaningful representations of cancer histology and transcriptomics. Coupled with clinical outcomes, these insights will offer an understanding of how tumor spatial organization influences disease progression and treatment response.

## Methods
### Human tissue samples
The institutional review board of the Lausanne University Hospital and the local ethics committee approved the study, in accordance with the Helsinki Declaration (CER-VD 2023-00080). Written informed consent was available for all patients. Tissue sections were obtained from archival FFPE tumor samples. Histological diagnoses were reviewed by consensus between two pathologists (K.v.L. and C.S. for Breast and Lung; L.d.L. and C.S. for DLBCL). Detailed information on histotype, block age, and DV200 values are listed in Supplementary Data 1.

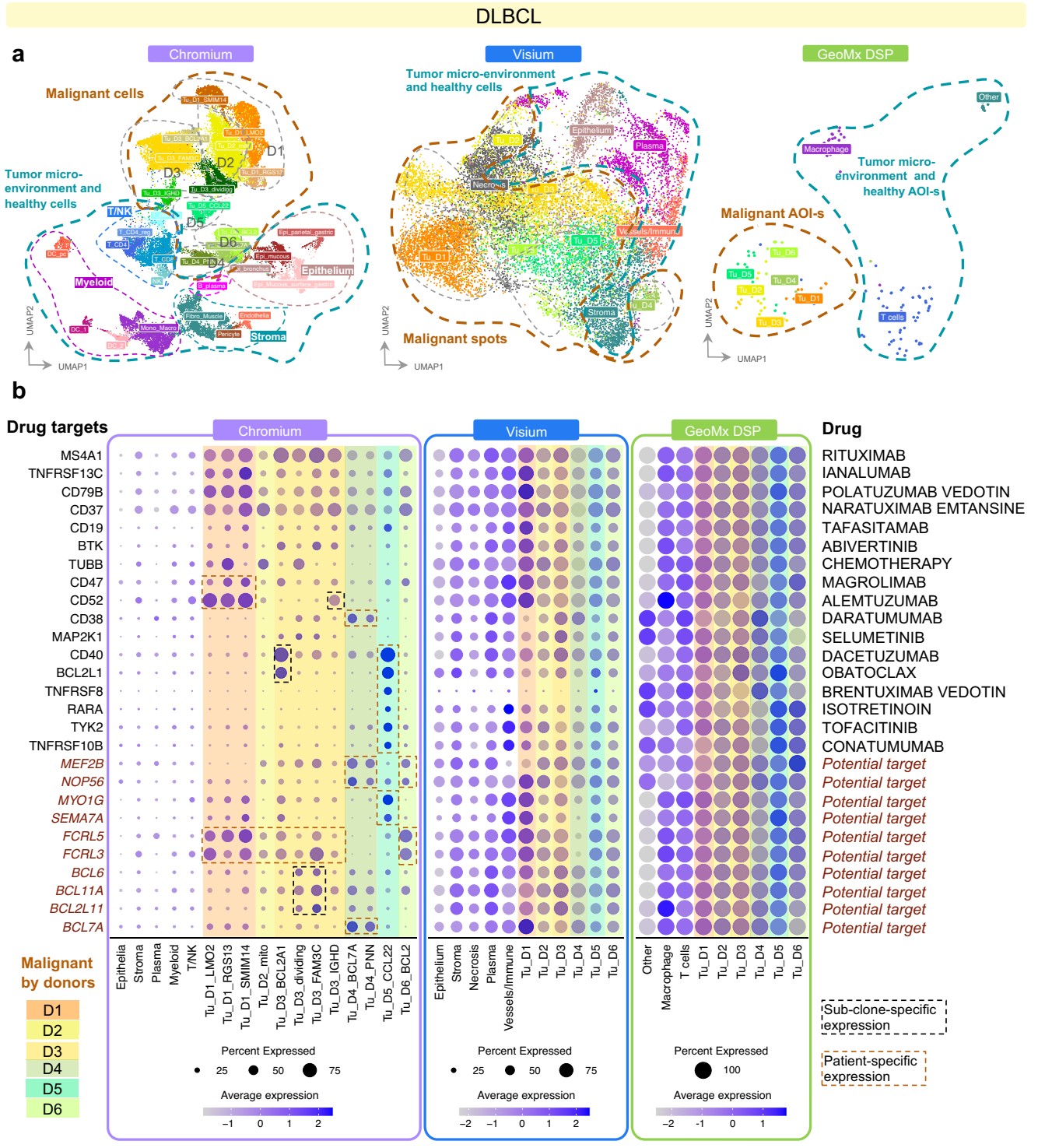

**Fig. 6 | Exploration of drug targets and patient subgroups in DLBCL. a** UMAP representations of Chromium, Visium and GeoMx DLBCL patients, colored by Level 4 cell types, clustering-based annotation of spots, or AOI types correspondingly. 6 samples, Chromium: 39,713 cells; Visium: 18,580 spots; GeoMx: 136 AOIs. **b** Expression of selected genes that are significantly differentially expressed in at least one donor subtype compared to TME cells in any of the three methods. Genes for which there are existing drugs (see further information in Supplementary Data 8) are listed per row. Genes with no drug are labeled as "Potential target". Gene expression in any of the donors is color-coded. Patient-specific and subclone specific gene expression is highlighted with dashed lines. Source data are provided as a Source Data file.

**Chromium FLEX**

**Lab workflow.** For the single-cell transcriptomics, two FFPE curls of 25 micrometers were sectioned from each of the FFPE block. Nuclei were extracted using snPATHO protocol[13], based on the protocol from 10X (CG000632). The nuclei were counted on the LUNA-FX7 cell counter (AO/PI viability kit, Logos). Pools of four samples were processed together as shown in Supplementary Data 4, targeting 40,000 nuclei recovery per pool (10,000 nuclei per sample). Replicas for two donors B3 and B4 were processed independently. The protocol was followed by the manufacturer's protocol Chromium Fixed RNA profiling - Multiplexed samples (CG000527). Libraries were sequenced using a NovaSeq™ 6000/X sequencer aiming for 10,000 read pairs/nuclei.

**Table 1 | Comparison of the methods**

| | Task | GeoMx DSP Whole Transcriptome ST | Visium CytAssist ST | Chromium scRNAseq |
|---|---|---|---|---|
| **OVERVIEW** | Imaging data | IF (3 markers + nuclear) | H&E or IF* | none |
| | Description of data unit | 25-200 cells pooled, pre-selected by IF marker | 1-50 cells pooled in a circle of 55 μm diameter | single cell, no spatial resolution |
| **EXP. DESIGN** | Input material | 5 μm section, surface up to 12 x 10 mm$^2$✳ | 10 μm (5 μm for DLBCL) section, surface up to 6.5 x 6.5 mm$^2$ | 1 x 25 μm curl |
| | Number of aimed data units/patient | 24✳ AOIs | Up to 4992 spots | 10,000 cells |
| **WET-LAB** | Workflow difficulty | Advanced: operator makes decisions for each individual data point. Requires optimisation of protocol for every tissue and combination of fluorescent markers. | Medium: follow manufacturer's protocol | Medium: follow manufacturer's protocol |
| | Cost | ~$300,000 Digital Spatial Profiler + ~$1500 per sample (increases with more AOIs) | ~$80,000 CytAssist + ~$2450 per sample | ~$65,000 Chromium + ~$750 per sample |
| | Max throughput✛ | 8-12 samples/week✳ | 16 samples/week | 64 samples/week |
| **DATA** | Computational resources | ~30min for the cohort starting from .dcc files✳ | ~1h for the cohort starting from .fastq files✳ | ~8h for the cohort starting from .fastq files✳ |
| | Data analysis | Easy-to-use software in the DSP machine. Limited tools available by the community, requiring more custom code for advanced analysis. | Easy-to-use user software Loupe Browser. Good selection of community-developed tools available. | Easy-to-use user software Loupe Browser. Good selection of community-developed tools available. |
| | Further comment | Access of files from the machine is slow and can take hours per slide for copying. Unexpected contamination appeared between the AOI-s. | Observed unexpected patient-effect | Probe-based data might not be suitable with all the previous tools made mainly on 3', 5' or full-length sequencing data |

**Table 1 (continued) | Comparison of the methods**

| | | Mixtures of cell types within ROIs and even within cell-type specific AOIs. | Mixtures of cell types within spots | Pure cell type transcriptomes |
|---|---|---|---|---|
| **D I S E A S E** | Cell type specificity | | | |
| | Spatial organization of cell types | Available in pre-defined ROIs | Available across the assayed tissue area | none |
| | Identification of localized structures (e.g. TLS) | Feasible within pre-selected AOI segments with reduced certainty. | Feasible across assayed tissue area. Computational enhancement (e.g., BayesSpace) improves certainty. | none |
| **B I O L O G Y** | DE genes detection for intra-tumor subtypes | Possible only across selected ROIs. DE genes consistent with Visium but decreased statistical power. | Possible across any region in the assayed area. Identifies highest number of distinct DE genes due to large number of spots that increase statistical power | Identifies DE genes consistent with Visium but spatial mapping needs coupling with spatial method. |
| | Drug target detection (informed by deconvolution) | Inter-patient heterogeneity remained ambiguous even when using Chromium data for deconvolution. | Enhanced clarity through deconvolution using Chromium data. Helps map targets spatially. | High resolution of drug target expression between patients and malignant subtypes within patient. |

*FTE* Full-time equivalent

*AOI* Area of Illumination

*ST* Spatial Transcriptomics

*scRNAseq* single-cell RNA sequencing

*IF* Immunofluorescence

\* Optional

‡ Total capture area is 12×24 mm, but in this experiment 3 sections were placed in the same area in order to save cost

⸫ User chooses the amount, correlated with cost

✳ Throughput is higher with less AOIs and segments

❖ 2 FTE-s and 1 instrument

❋ On a 16 CPU and 64Go of RAM machine

Grey (+) - positive

Grey (++) - neutral

Grey (+++) - negative

bcl2fastq was used for demultiplexing libraries after sequencing. FASTQ files were processed with Cell Ranger 7.1.0 (10X Genomics) with multi pipeline and human genome reference GRCh38-2020-A.

**Pre-processing, quality control (QC) and annotation.** Filtered barcode counts matrices from CellRanger were imported into R (v. 4.3.2, https://www.R-project.org/) analyses by the Seurat package (v. 5.0.1)[19]. *SoupX*[20] was used to remove ambient RNA contamination. Only cells with minimum 200 genes (and max 6000 genes in DLBCL only), and < 10% (and < 20% in breast and lung) of reads mapping to mitochondrial genes were kept. For each sample, corrected raw counts were normalized using *SCTransform*. The top 3000 variable genes across samples were selected using the *SelectIntegrationFeatures* function. Dimensionality reduction using principal component analysis (PCA) was done, followed by a Uniform Manifold Approximation and Projection (UMAP) dimensional reduction using 50 principal components. Clustering with shared nearest neighbor (SNN) modularity optimization-based clustering algorithm implemented in the *FindNeighbors()* and *FindClusters()* functions wereas performed, with 30 principal components and resolutions between 0.4 and 0.8. The expression level of canonical marker genes and the top differentially expressed genes were used for identifying known cell types corresponding to the clusters. We finally performed sub-clustering to increase the granularity of annotations.

**Differential expression (DE) analysis.** We used *Seurat::FindAllMarkers()* to find significantly (Wilcoxon's rank sum test, two-sided, multiple comparisons corrected with default Bonferroni method, adjusted *p* value < 0.05, log fold change > 1) differentially expressed genes between clusters of interest.

**Deconvolution.** Chromium data are used as reference for deconvolution with cell type annotation at different levels of resolution. Since breast and lung cancer samples are both solid tumor and are

annotated together, when constructing Chromium reference, we combine healthy cell types from both indications and add indication-specific tumor cells at Level 4.

For Fig. 2 and Fig. 3 on deconvolution cell type specificity of GeoMx and Visium, we used a combination of Level 1 and Level 4 annotation as Level 1.5 so that we obtain T cells and Macrophage cell types of interest while keeping concise grouping of other healthy or tumor cell types. For breast and lung cancer, T cells cell type is composed of "T_CD4", "T_CD8_exhausted", "T_CTL", "T_CXCL13", "T_reg", and "TNK_dividing" from Level 4 annotation. For DLBCL, "T_CD4", "T_CD4_reg", "T_CD8", "T_dividing" are grouped as T cells. For breast and lung cancer, we grouped Macrophage and Monocyte from Level 4 annotation as Macrophage. Because in GeoMx, the marker used for staining the *Macrophage* region is CD68 +, which is expressed in both macrophage and monocyte cell types. For DLBCL, "Mono_Macro" cell type is labeled as Macrophage. Myeloid else consists of "DC_1", "DC_2", "DC_activated", "DC_pc", "Granulocyte", "Mast_cell" for breast and lung, and "DC_1", "DC_2", "DC_pc" for DLBCL.

For Figs. 4, 5 on Visium enabling biology discovery, to show scatterpie plot of deconvolution result for each patient and to obtain the deconvolution cell type majority vote for each spot, the tumor signature in Chromium reference was filtered to each patient's Level 4 tumor subtypes, if there are multiple subtypes.

**Intra- and inter-patient tumor heterogeneity analysis.** For Fig. 6, our interest is to identify the expression level of drug targets in tumor cell types in Chromium. Therefore, we make a condensed aggregation of healthy cell types from Level 1 annotation, while separating out tumor subtypes within each patient, if there are multiple subtypes, by adopting Level 4.

In Supplementary Fig. 15, to borrow insights from Chromium and enhance the drug discovery capability of Visium and GeoMx, healthy cell types at Level 1 and patient-specific tumor cells from Level 2 annotation are used for Chromium. For Visium, the density of deconvolution fraction for each targeted cell type in the corresponding majority voted cell type category were grouped from Level 4 to Level 1 for healthy spots and Level 2 for tumor spots. Similarly for GeoMx, the density of deconvolution fraction for each targeted cell type in the corresponding AOI labeled segments were combined in the following way: cell type "Other" contains "Plasma", "Epithelia", "Fibro_muscle", and "Vessel"; cell type "T cells" consists of "T-cell" and "NK"; cell type "Macrophage" is characterized as "Myeloid".

Known and potential drug targets were selected manually from a genes list that showed significantly higher expression in any technologies for malignant cells/regions/AOI label versus non-malignant cells/regions/AOI label. Gene expression specificity for malignant cells was considered, as well as consultation of literature, for selecting potential drug targets. Gene-drug matches in Supplementary Data 8, that were used in Figs. 5, 6, and Supplementary Figs. 11, 12 were collected from the ChEMBL database[21]. DE analysis on drug targets was performed separately on Chromium, Visium and GeoMx data. Fold changes and *p*-values obtained using Wilcoxon rank sum test (FindMarkers function from Seurat v. 5.0.1) for all the genes were calculated per donor per technology between malignant cells, regions or AOI labels versus non-malignant cells, regions or AOI labels in Chromium, Visium and GeoMx correspondingly. The gene rank was determined by the *p*-value among all expressed genes per donor per technology. The statistics' are compiled in Supplementary Data 9, and visualised in Supplementary Fig. 14.

**GeoMx DSP**
**Lab workflow.** All Breast, Lung and DLBCL samples were assayed on the Nanostring GeoMx Digital Spatial Profiler (DSP) platform using the Whole Transcriptome Atlas (WTA) probe panel with NGS readout. Two to three tissue sections were placed on each slide (Supplementary Fig. 1a). Two samples (one each for Breast and Lung, B1 and L1 respectively) were analyzed in duplicate. Due to the failure of the GeoMx DSP instrument, two samples were repeated, including L4 and a duplicate of L3.

Slides were processed according to the manufacturer's instructions. Tissue sections were cut at 5 um thickness. They were baked at 60 °C for 2 hours. Sections were stained on a BOND-RXm fully automated multiplexing immunohistochemical stainer (Leica Biosystems; Wetzlar, Germany) using the following immunofluorescence antibodies: for Breast and Lung, PanCK - AF532 (Clone AE1 + AE3 from Novus) diluted at 1/50, CD3 - AF647 (Clone CD3-12 from Biorad) diluted at 1/100, CD68 - AF594 (Clone KP1, from Santa Cruz) in concentration 0.25 ug/ml and SYTO 13 nuclear stain diluted at 1:10000 (Nanostring) in Buffer W for 1 h at room temperature; for DLBCL, CD20 - AF594 (Clone IGL/773 from BioTechne) diluted at 1/100, CD3 - AF647 (Clone CD3-12 from Biorad) diluted at 1/100, CD68 - AF488 (Clone KP1 from e-BioMed) at concentration 0.25 ug/ml and SYTO 83 nuclear stain diluted at 1:100000 (Invitrogen) in Buffer W for 1 h at room temperature.

Regions of interest (ROIs) were selected by pathologists, and segmented into areas of illumination (AOI-s) based on immunofluorescent markers (Supplementary Datas 3-4). The first segment was defined as double positive for CD3 and CD68 with the purpose of excluding autofluorescent structures (such as elastin fibers in lung) or debris, and it was not collected. For Lung and DLBCL samples, the CD3-*AOI-s* were collected first, followed by CD68-*AOI-s*. For Breast samples, CD68-*AOI-s* were collected first, followed by CD3-*AOI-s*. PanCK-*AOI-s* and CD20-*AOI-s* were collected third for solid cancers and DLBCL respectively. For five and four ROI-s in breast and lung replica samples respectively, only two segments of PanCK+ and PanCK- were collected in that order, ignoring other markers. The remaining cells showing SYTO nuclear expression in the absence of any cytoplasmic marker were collected as the fourth marker-negative (*Other*) segment in BC. All the remaining surface of ROIs which had not been collected into any AOI was collected as marker-negative (*Other*) segments for Lung and DLBCL, independently of the presence of nuclei. With the exception of the *Other* segment of Lung and DLBCL, a segment was collected when it contained at least 50 cells. About 24 such AOI-s per patient were collected with AOI size varying from about 500 $\mu m^2$ [2] to 300,000 $\mu m^2$ with an average size of 60,000 $\mu m^2$. Library preparation and sequencing was performed according to the manufacturer's instructions and kits. Briefly, i5 and i7 indices were added, reactions pooled and purified, and libraries sequenced with paired-read sequences with 2 × 27 base pairs. Manual curation of the annotation file was performed to match the collected segments with a biological cell fragment as AOI label (Supplementary Data 3).

**Pre-processing and QC.** The GeoMx NGS pipeline *GeomxTools*[22] was used to convert FASTQ files into expression matrices of raw probe counts stored in DCC files. AOI segment QC was conducted using NanoString recommendations: segments with > 1000 raw reads, < 80% aligned, trimmed or stitched, < 50% saturation, > 1000 NTC and < 100 nuclei are removed. There were 2 breast segments removed due to low saturation, 1 lung segment excluded due to low alignment, and 7 DLBCL segments were filtered out due to low reads ($n = 1$), low saturation ($n = 2$), and small nuclei area ($n = 4$). This results in a sample size of 122, 117, and 137 for breast, lung, and DLBCL samples, respectively.

Probes with geometric mean from all segments divided by the geometric mean of all probe counts representing the target from all segments less than 0.1, together with probes flagged as outliers according to the Grubb's test in at least 20% of the segments, were filtered out. Gene raw counts were generated using the geometric mean of the associated probe counts. Meta variable number of genes detected were derived and plotted in Fig. 1d GeoMx panel.

Segments from each indication were combined by *GeomxTools* as a *SummarizedExperiment* object, and then converted by *standR*[23] to a *SpatialExperiment* object. No AOI segment or gene was further excluded with the *standR* preprocessing pipeline.

**Normalization and batch correction.** As suggested by van Hijfte et al[24]., we use quantile normalization for GeoMx data analysis. Quantile normalization (*preprocessCore::normalize.quantile()*) was applied to *log1p()* transformed data. The reduced dimension of normalized GeoMx breast samples is shown in Fig. 1d. We applied batch correction method RUV4, as implemented in *standR*, to quantile normalized data to remove batch effects introduced by slides. Distinct *T cells* and *Macrophage* clusters on the reduced dimension of normalized and batch corrected data are shown in Supplementary Fig. 7d. The corrected batch effect is shown in Supplementary Fig. 12.

**Reduced dimensions.** We used R package *scater*, *runPCA()*, *runUMAP()*, and *runTSNE()* to obtain reduced dimensions. For the GeoMx data, we used t-SNE embedding for visualization due to the small number of data points, as it provided a clear and not cluttered representation.

**DE analysis.** Given GeoMx data closely resembles bulk RNA-seq, we used the *limma-voom*[25] pipeline, by using *voom()*, *lmFit()*, *contrasts.fit()* and *eBayes()* functions, as suggested in *standR*, to find significantly (moderated *t*-test, two-sided, multiple comparisons corrected with default Benjamini-Hochberg method, adjusted *p* value < 0.05, log fold change > 1) differentially expressed genes between clusters of interest.

**Deconvolution.** Cell type abundance was estimated using the *SpatialDecon* package[14] and the signature matrix derived from the Chromium single-cell RNAseq dataset. When running deconvolution, we exponentiated back the log transformed, quantile normalized, and RUV4 batch corrected assay to obtain normalized and batch corrected data on linear scale, as suggested by *SpatialDecon::spatialdecon()* manual. To enhance the drug discovery potential of GeoMx, we borrowed insights from Chromium through deconvolution as shown in Supplementary Fig. 15d, e.

**Visualization.** Deconvolution cell type proportions and marker gene expressions across all AOIs in each ROI were visualized spatially with R package *SpatialOmicsOverlay*[26]. The fluorescence image, ROI pathology class and AOI segmentations were visualized in QuPath[27].

### Visium CytAssist
**Lab workflow.** Sections with 10 micrometers for lung and breast, and 5 micrometers for DLBCL were placed on individual positively charged slides (Superfrost Plus Adhesion Microscope Slides, thermofisher). The slide preparation workflow was performed using the Visium CytAssist Spatial Gene Expression for FFPE by 10X (CG000520 and CG000518). High resolution H&E imaging was done on Aperio CS at 40x magnification. The 6.5 mm × 6.5 mm capture area was chosen by the pathologists. Manufacturer's workflow was followed until the library preparation. Libraries were sequenced with paired-end dual-indexing (28 cycles Read 1, 10 cycles i7, 10 cycles i5, 90 cycles Read 2) on NovaSeq™ 6000/X. Space Ranger 2.0.1 pipeline was used to align the reads to human genome reference GRCh38-2020-A. High resolution image and CytAssist image were aligned using Space Ranger.

**Manual annotation of the Visium spots.** Loupe Browser version 6.5.0 was used by the pathologists to annotate all of the Visium spots in one of the following classes: pure tumor (invasive carcinoma), in situ carcinoma, tumor-stroma mix, intratumoral stroma, lymphocytes, immune cell mix, tumor infiltrating lymphocytes, intratumoral vessels, artefact/fold/empty (for spots to be excluded from the analysis due to sectioning artefacts or falling outside the tissue), most likely tumor (for epithelial proliferations which were difficult to classify based on H&E staining alone), acellular mucin, necrosis/debris for solid Breast and Lung cancer; tumor, small lymphocytes, stroma, necrosis, epithelium, and vessels for DLBCL. Visium spots, each covering a mixture of cells, were assigned to one of the previously listed classes according to the majority cell type (> 50%) observed on H&E staining. The hybrid category "tumor-stroma mix" was used for spots covering an approximately equivalent mixture of tumoral cells and stroma. Immune cell mix annotation class was used for spots covering a mixture of cells, including macrophages, lymphocytes, and plasmocytes (Supplementary Data 5, Supplementary Fig. 2c).

**Pre-processing and QC.** Using the filtered gene count matrix, we excluded spots that had fewer than 100 unique molecular identifiers (UMIs), more than 22% mitochondrial reads, or were located at the edge of the fiducial frame, unreasonably distant from the majority of the tissue. Additionally, spots identified by pathologists as artifacts, folds, or empty were also filtered out. We set a quality control threshold requiring genes to be detected in at least 20 spots to be included in the analysis.

**Deconvolution.** After QC, cell type fraction is estimated using deconvolution method *Cell2location*[15], which takes raw unique molecular identifier (UMI) counts of each Visium sample and annotated single cell reference data as input. We defined a Visium annotation approach from deconvolution majority vote with Level 4 Chromium annotation, by setting a cut-off that if the highest cell type fraction is above 20%, we label the spot as that cell type, otherwise, we label the spot as "Mix". To enhance the drug discovery potential of Visium, we borrowed insights from Chromium through deconvolution as shown in Supplementary Fig. 15b, e.

**Normalization.** We adapted the recommended normalization method for different biological questions. To show sample reproducibility among replicates and to integrate samples within each indication, we followed the Seurat workflow[19] and used *SCTransform* normalization. The reduced dimension of normalized Visium breast samples is shown in Fig. 1d Visium panel. On the other hand, we used the default log normalization for individual sample's spatial aware clustering (Fig. 5b) to achieve optimal spatial domain detection that detects intra-patient heterogeneity, as recommended in the BayesSpace vignette.

**Clustering.** For individual samples (Fig. 5b), we used the graph-based clustering approach implemented in Seurat to identify the optimal number of clusters for *BayesSpace*[16]. We followed the default *BayesSpace* pipeline and parameters, with log normalization and 50 PCs. The log transformed subspot gene expression was inferred from subspot level PCs, with a linear model projection between spot level log transformed gene expression to spot level PCs, as detailed in *BayesSpace*.

**DE analysis.** Given Visium can have nearly single-cell resolution, we used *Seurat::FindAllMarkers()* to find significantly (Wilcoxon's rank sum test, two-sided, multiple comparisons corrected with default Bonferroni method, adjusted *p* value < 0.05, log fold change > 1) differentially expressed genes between clusters of interest.

**Integration.** To integrate multiple samples with the same indication across patients, we decontaminated Visium counts with SpotClean[28] and used SCTransform normalization for each sample (Supplementary Fig. 10, 11). *SpotClean::spotclean()* is run with the default parameter on the raw gene count matrix from the SpaceRanger output. We then performed QC of the spot-cleaned object by filtering to the gene and spot barcodes that passed QC previously. We used *Seurat::SCTransform()* in Seurat (v.5.0.1)[19] to normalize each spot-cleaned Visium

sample and to identify variable genes. We used *Seurat::SelectIntegrationFeatures()* function to select a union of 3000 HVGs by consensus ranking of the gene from all samples. Standard Seurat pipeline (e.g., *RunPCA()*, *FindNeighbors()*, *FindClusters()*) is used to obtain the reduced dimension UMAP and clusters.

DLBCL Visium data was annotated into regions after Seurat clustering on the integrated object (Supplementary Fig. 13a). Donor and pathology annotation distribution was considered for each cluster to annotate clusters 4 (D5) and 5 (D6) and clusters 2, 12, 14, 16 (necrosis) correspondingly (Supplementary Fig 13b, c). Canonical marker genes' expression in each cluster (Supplementary Fig. 13d) aided to annotate regions: clusters 9, 10 and 13 were mostly expressing plasma cell marker genes, cluster 6 was assigned to epithelium, and cluster 1 was showing strong stroma signal. Cluster 7 was likely a mixture of immune cells with vessels, which was confirmed by pathology annotation. Aggregated patient-specific tumor markers from Supplementary Fig. 4 were used to annotate malignant areas: cluster 0 (D1), clusters 15, 18 (D2), clusters 3, 8, 11 (D3), and cluster 17 (D4) were identified. *Seurat::FindAllMarkers()* was further consulted for differentially expressed genes between clusters.

**Visualization.** Deconvolution cell type proportion across all spots in one sample, pathology annotation, clusters and other continuous values, such as gene expression and a single cell type deconvolution proportion, as well as reduced dimensions were visualized spatially with R packages *Seurat*[19] and *ggspavis*[29].

**Spot-segment matching between Visium and GeoMx.** Fluorescent images carrying GeoMx's regions of interest and areas of interest were registered to high resolution H&E images carrying Visium spots using the Elastix software[30,31]. For GeoMx images, registration was performed on the SYTO 13 channel, with pixel intensities clamped to 1% and 99% of extremal values in order to alleviate fluorescent artefacts. H&E images were converted to grayscale prior to registration. Registration was performed on downsampled images with a resolution of (approximately) 4 microns per pixel. Affine registration parameters were optimized by minimizing Mattes advanced mutual information metrics[30]. As GeoMx's fluorescent image and Visium's H&E were carried on two subsequent slides of the same block, a visual inspection was performed to ensure the quality of the registration procedure. For mapped spots and subspots, PanCK-AOI-s are excluded from the analysis due to their impurity by definition.

**Statistics and reproducibility.** Four patients in Breast and NSCLC, and 6 patients in DLBCL were profiled. Two patients in Breast and Lung were repeated with GeoMx and Visium, and two Breast patients were repeated in Chromium. H&E staining was performed once. Figure panels on Figs. 4 and 5, and Supplementary Fig. 9 demonstrate specific structures and were found on specific donors.

### Reporting summary
Further information on research design is available in the Nature Portfolio Reporting Summary linked to this article.

## Data availability
The Visium, Chromium and GeoMx raw and processed data generated in this study have been deposited in the ArrayExpress database under accession codes E-MTAB-14560 and E-MTAB-14566. In addition, processed single-cell Chromium data are available for browsing at cellxgene database [https://cellxgene.cziscience.com/collections/bd552f76-1f1b-43a3-b9ee-0aace57e90d6]. The remaining data are available within the Article, Supplementary Information or Source Data file. Source data are provided with this paper.

## Code availability
The source code of the MOSAIC pilot manuscript is available at https://github.com/bdsc-tds/mosaic_pilot_study[32]. Instructions on how to reproduce the analysis are specified in the README file. Specific software and package versions are listed as a singularity container script and a conda environment file in the renv.lock file and env.yml file, respectively, in the repository. Key packages are: R version 4.3.2, Bioconductor v3.18, Seurat v5.0.1, BayesSpace v1.11.9, ggplot2 v3.5.1, ggspavis v1.3.1, GeomxTools v3.5.0, SpatialDecon v1.12.0, SpotClean v1.4.1, dplyr v1.3.1, scran v1.30.2, ComplexHeatmap v2.18.0, SpatialOmicsOverlay v1.2.1, magick v2.8.2. Python version 3.9.18, cell2location v0.1.3, squidpy v1.3.0.

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

## Acknowledgements

Data generation of this work was funded by Owkin. Part of this work was supported by SNSF grant #320030_215550 to R.G. We thank 10x Genomics and Nanostring for valuable discussions. We thank Jonathan Thevenet for contributions to data generation.

## Author contributions

Y.D., C.S., Q.B., A.E.P. and S.C. all contributed to the exploration of the data and all of these authors contributed equally. Y.D. developed the pipelines for Visium and GeoMx analyses, generated, designed, and vectorized all the figures, drafted the result section of the paper, facilitated submission and performed revision, and maintains the GitHub repository. Q.B. contributed results for Visium deconvolution, and generated initial figures. A.E.P. developed an alternative GeoMx pipeline for exploration and contributed to marker gene rank statistics in revision. S.C. contributed to the Visium integration method and disease biology insights. S.T., A.K., S.L.L, K.v.L., C.Ho, R.G, K.H. and E.M. prepared the experimental design. N.P., C.S., and L.d.L. contributed to sample access. D.B., E.M. and S.C. annotated Chromium data. S.T., N.P., C.S, K.v.L., and E.M. generated the GeoMx data. S.A. and M.A.G. generated the Chromium and Visium data. R.D. and R.San. performed head-to-head registration of Visium and GeoMx, and programmed scripts to display GeoMx metadata in QuPath. S.K. uploaded data to ArrayExpress and performed data operations. C.Ha. managed collaborations and data production and analysis processes. R.Sar. produced Fig. 1a and facilitated data generation. K.V.L and C.S. annotated Visium spots. C.S, Q.B., A.E.P, S.C. S.L.L, N.P., E.Y.D., L.d.L., R.G. and E.M. prepared the first version of the results. A.K., C.Ho. contributed to drug target literature review and drafted the related method section. G.C., S.P., V.S, C.Ho., K.H and L.d.L. provided medical context, interpretation and access to the samples. R.G. & E.M. drafted the introduction and discussion of the paper, proposed ideas during analysis and paper revision, and reviewed the paper drafts. L.D.L., R.G., K.H., and E.M. jointly supervised this work. All of the authors contributed to the interpretation of the data.

## Competing interests

E.M., A.E.P., Q.B., S.C., E.Y.D., R.San., K.v.L., R.D., C.Ha., S.L.L., A. K., V.S. are Owkin employees and shareholders. C. Ho. is an Owkin employee and a consultant for Nanobiotix. R.G. has received consulting income from Takeda, Arcellx, GSK, and Sanofi; declares ownership in Ozette Technologies; and has received research funding from 10X Genomics through his employer, the CHUV. G.C. has received honoraria from Bristol-Myers Squibb. The Lausanne University Hospital (CHUV) has received honoraria for advisory services G.C. has provided to Iovance and EVIR. G.C. has received royalties from the University of Pennsylvania for CAR T cell therapy licensed to Novartis and Tmunity Therapeutics, and from the Ludwig Institute for Cancer Research, the University of Lausanne and the CHUV, for NeoTIL intellectual property previously licensed to Tigen Pharma. S.P. has received educational grants, provided consultation, attended advisory boards, and/or delivered lectures for the following organizations, from which S.P. has received honoraria (all fees directed to their institution): AbbVie, Amgen, Arcus, AstraZeneca, Bayer, Beigene, BioNTech, BerGenBio, Bicycle Therapeutics, Biocartis, BioInvent, Blueprint Medicines, Boehringer Ingelheim, Bristol-Myers Squibb, Clovis, Daiichi Sankyo, Debiopharm, Eli Lilly, F-Star, Foundation Medicine, Genmab, Genzyme, Gilead, GSK, Hutchmed, Illumina, Incyte, Ipsen, iTeos, Janssen, Qlucore, Merck Sharp and Dohme, Merck Serono, Merrimack, Mirati, Nuvation Bio, Nykode Therapeutics, Novartis, Novocure, Pharma Mar, Promontory Therapeutics, Pfizer, Regeneron, Roche/Genentech, Sanofi, Seattle Genetics, Takeda, and Zymeworks. S.P. has spoken at company-organized public events for AstraZeneca, Boehringer Ingelheim, Bristol-Myers Squibb, Eli Lilly, Foundation Medicine, GSK, Illumina, Ipsen, Merck Sharp and Dohme, Mirati, Novartis, Pfizer, Roche/Genentech, Sanofi, Seattle Genetics, and Takeda. Additionally, S.P. has served as a principal investigator for trials sponsored by Amgen, Arcus, AstraZeneca, Beigene, Bristol-Myers Squibb, Eli Lilly, GSK, iTeos, Merck Sharp and Dohme, Mirati, Pharma Mar, Promontory Therapeutics, Roche/Genentech, and Seattle Genetics, with institutional financial support for these clinical trials.
