## [Transparent Peer Review file · Nature Communications]

Transcriptome Analysis of Archived Tumors by Visium, GeoMx DSP, and Chromium Reveals Patient Heterogeneity

Corresponding Author: Professor Raphael Gottardo

Version 0:

Reviewer comments:

Reviewer #1

(Remarks to the Author)

In this manuscript, Dong et al. conducted a comparative study on two spatially resolved full-transcriptomics methods, GeoMX and Visium, using Chromium FLEX to provide single-cell-level transcriptomic data as a reference for the necessary analysis. The authors analyzed samples from 14 patients across all three methods in parallel, assessing cell type signature specificity between the two spatial methods.

The authors are commended for their effort in conducting a comparative study of two spatial transcriptomics methods. However, a primary concern is that some analyses focus solely on Visium data without a direct comparative context, potentially detracting from the study's novelty and alignment with its core objective. Figures 4 and 5, along with three paragraphs in the Results section, lack a clear purpose and main conclusions relevant to the study's central focus. This reviewer has significant concerns regarding the novelty of this portion.

Specific Comments:

1. The results in Figure 4 used a deconvolution method to conclude the rich information that spatial transcriptomic data can offer. This topic has been extensively benchmarked (see, e.g., <https://doi.org/10.1038/s41467-023-37168-7>). Many of these deconvolution methods, along with prior benchmarking studies, have effectively demonstrated their value in enhancing spatial data annotation. As such, this section does not directly contribute to the comparative analysis of spatial methods, which is the primary focus of the study. This raises concerns regarding the aim of this section.
2. Similarly, the results presented in Figure 5 used the same deconvolution approach, only to confirm findings from previous method on sub-resolution annotation models for Visium data, rather than contributing new insights. While this validation may be informative, it lacks direct relevance to the study's comparative goal of evaluating two distinct spatial transcriptomics methods.

(Remarks on code availability)

Reviewer #2

(Remarks to the Author)

In this paper, Dong et al. conducted a comparative analysis among probe-based full transcriptome platforms, Visium, Chromium and GeoMX, and summarized advantages and disadvantages, focusing on experimental challenges, bioinformatics approaches and the way to interpret cancer biology. The authors analyzed lung cancer, breast cancer and B-cell lymphoma specimens. They described features of each platform and introduce suitable computational approaches for each dataset. However, it is not clear how to make platform decisions. The biological findings which were identified in this study was also not sufficiently described.

Points;

1. Table 1 is useful for readers but lengthy. It is still not clear how should we choose platforms.
2. In Fig. 1b, DV200 of breast mucinous carcinoma is quite low. Did the RNA quality affect the data quality? The authors should add the discussion about the relationship between sample and data quality to Fig. 1b. Sample ID can be also added to Fig. 1b.

3. There is no sufficient description and discussion for Fig. 1c. The reviewer wondered why high variation in the number of detected genes were observed (technical batch effects, biological differences, etc. in addition to platform differences).
4. Why was the t-SNE plot used for GeoMX DSP visualization in Fig. 1d? The authors used UMAP plots for Visium and Chromium.
5. The authors should indicate numerically or statistically the extent of overlap in comparison between pathology label and deconvolution in Fig. 4a and b.
6. In Fig. 5 and 6, the authors showed several potential therapeutic targets and drugs. The reviewer do not know whether these targets and drugs have been validated and approved. The detailed description and discussion should be needed.
7. The authors successfully captured the intra-patient heterogeneity using Chromium or Visium. However, they did not sufficiently mention how we can evaluate these information and use them to identify targeted therapies.
8. The reviewer recommend that the authors would include the discussion about the previous studies (e.g. Janesick et al. 2023 Nat Commun; PMID: 38114474) in the context of the comparative analysis of different platforms.

(Remarks on code availability)

The reviewer has just visited the github page and confirmed there are enough contents. The reviewer did not install and run the code because the reviewer had no time to do.

Reviewer #3

(Remarks to the Author)

This study conducts a technical comparison of the validity of Visium CytAssist, Chromium Flex, and GeoMx DSP on FFPE samples of different tumors. While such technical comparison is of broad interest, the conclusions are expected. In other words, the current manuscript brings new data to the scientific community but does not bring new insights or knowledge.

(Remarks on code availability)

Reviewer #4

(Remarks to the Author)

(Remarks on code availability)

Version 1:

Reviewer comments:

Reviewer #1

(Remarks to the Author)

In the revised manuscript, the authors have adequately addressed my comments. Specifically, the results related to Figures 4 and 5 have been updated to strengthen the comparison between the Visium and GeoMX methods. These revisions enhance the novelty of the study and provide insights into platform characteristics, aiding in informed decision-making.

(Remarks on code availability)

Reviewer #2

(Remarks to the Author)

The authors made the effort to add information and descriptions which the reviewer pointed out to the revised manuscript. While the Discussion section was improved, there are still lacks of indication of novelty. For example, in Table 1, if the authors focused on technical comparison in this paper, they should clearly and simply indicate platform differences which were identified by their own study. The current Table 1 seems to be Supplementary information because there are less novel information. The authors should provide a detailed summary of the molecular/cellular events that can only be identified using each platform.

The reviewer could not also clearly understand how we can use the information on intra-tumor heterogeneity to identify targeted therapies. Heterogeneity is an important information on single-cell/spatial transcriptome data. If DE genes in

specific clusters were used as target genes, tumor cells would be expected to be only partially eliminated. Should we use common genes among clusters (patient-specific expression) as target genes?

(Remarks on code availability)

Reviewer #3

(Remarks to the Author)

The authors have improved their manuscript extensively and brought new findings in addition to comparisons. It is acceptable now in my opinion.

(Remarks on code availability)

Reviewer #4

(Remarks to the Author)

(Remarks on code availability)

The code is organized with a README file providing instructions for installation and reproducibility.

Response to the reviewers

We sincerely appreciate the reviewers' thoughtful feedback, which has provided valuable insights to improve our work. We understand the reviewers' concerns and the need for substantial revisions to strengthen our study. After carefully reviewing their comments, we are confident that we have comprehensively addressed the points raised. Specifically:

1. We have significantly refined and enhanced our analysis to provide a more in-depth comparison between Visium and GeoMx, particularly in Figures 4 and 5 (Reviewers 1 and 2).
2. We have further clarified the biological findings within the context of the different platforms and consolidated our message and recommendations (Reviewers 2 and 3).
3. We have clearly articulated the novelty of our approach in the context of existing published work, emphasizing the importance of our data (Reviewer 3).

The manual nature of GeoMx data (selected ROIs) made this an extensive and labor-intensive effort, particularly for point 1, which we hope the referees will appreciate. This also highlights the inherent challenges of using GeoMx technology for large-scale studies.

Below, you will find our point-by-point response, where the reviewers' comments are presented in *italic*, followed by our responses in blue. We appreciate the opportunity to revise our manuscript and thank you for guiding us through this process. All changes have been highlighted in yellow in the revised manuscript.

Reviewer #1

In this manuscript, Dong et al. conducted a comparative study on two spatially resolved full-transcriptomics methods, GeoMX and Visium, using Chromium FLEX to provide single-cell-level transcriptomic data as a reference for the necessary analysis. The authors analyzed samples from 14 patients across all three methods in parallel, assessing cell type signature specificity between the two spatial methods.

The authors are commended for their effort in conducting a comparative study of two spatial transcriptomics methods. However, a primary concern is that some analyses focus solely on Visium data without a direct comparative context, potentially detracting from the study's novelty and alignment with its core objective. Figures 4 and 5, along with three paragraphs in the Results section, lack a clear purpose and main conclusions relevant to the study's central focus. This reviewer has significant concerns regarding the novelty of this portion.

Thank you for raising these points. We have addressed the questions in the following paragraphs. Specifically, we have now included direct comparative analyses for all figures. We acknowledge that this can be challenging due to the subjective and manual nature of GeoMx, where ROIs are selected visually. Nevertheless, we have made every effort to provide direct comparisons wherever possible.

Specific Comments:

The results in Figure 4 used a deconvolution method to conclude the rich information that spatial transcriptomic data can offer. This topic has been extensively benchmarked (see, e.g., <https://doi.org/10.1038/s41467-023-37168-7>). Many of these deconvolution methods, along with prior benchmarking studies, have effectively demonstrated their value in enhancing spatial data annotation. As such, this section does not directly contribute to the comparative analysis of spatial methods, which is the primary focus of the study. This raises concerns regarding the aim of this section.

Thank you for bringing this to our attention. We agree that the deconvolution methods add significant value, and acknowledge that our original analysis focused solely on Visium data. In response to this comment, we have updated Figure 4 in the manuscript by incorporating deconvolution results for GeoMx data as well, providing a direct comparison to support the primary focus of the study (see from page 9 lines 275-353).

We have rewritten this section to emphasize the comparative insights gained from deconvolution on both platforms. The content has shifted from highlighting the added value of deconvolution for Visium to a balanced comparison of the insights obtained using both methods. Our analysis demonstrates a high consistency of results in matching regions between Visium and Chromium. However, we note the sparse tissue coverage in GeoMx, which results in missing regions and structures, and high variation in cell type composition between AOI-s of the same segment. Additionally, we highlight the higher cell type-specific abundance in GeoMx as expected due to the segmentation approach, as well as contamination from neighboring cells in the AOI. These observations align with and support our main conclusions.

To our knowledge, this study provides the first head-to-head comparison of Visium and GeoMx using deconvolution on matched regions. This has also been highlighted in the introduction (see page 3, lines 107-111).

Similarly, the results presented in Figure 5 used the same deconvolution approach, only to confirm findings from previous method on sub-resolution annotation models for Visium data, rather than contributing new insights. While this validation may be informative, it lacks direct relevance to the study's comparative goal of evaluating two distinct spatial transcriptomics methods.

Our initial goal was to demonstrate that Visium data could be enhanced to improve sensitivity and resolution, addressing a common criticism of this platform. However, we have now completely revised the figure to include additional GeoMx data, providing a direct, head-to-head comparison of the two approaches.

Our results indicate that due to the selective and targeted nature of GeoMx, the TLS (tertiary lymphoid structure) is difficult to detect, even though the region was part of a selected ROI. The absence of specific B-cell markers/AOIs and the poor resolution of GeoMx make it difficult to confirm the presence of a TLS based on these data alone. In contrast, Visium data clearly highlight markers associated with TLS structures, a finding corroborated by the H&E

staining. This underscores our conclusion that GeoMx can introduce selective bias, potentially leading to the exclusion of critical immunological regions.

To address this, we have added a paragraph accompanying updated Figure 5, discussing these points (see page 12, lines 373-397) and included a supplementary figure (Supplementary Figure 9) showcasing another example: an immune-dense region (rich in B cells, T cells, and macrophages) that is challenging to dissect with GeoMx compared to Visium. These additions further reinforce the importance of Visium in capturing, in an unbiased fashion, the complexity of the immune microenvironment.

Additionally, we incorporated results from GeoMx into the intra-patient heterogeneity discovery panels as an addition to Figure 5. Similarities were observed with Visium and Chromium in detecting top differentially expressed genes between the two regions (see page 13, lines 399-448). Importantly, the heterogeneity was independently confirmed by all three technologies.

Reviewer #2

In this paper, Dong et al. conducted a comparative analysis among probe-based full transcriptome platforms, Visium, Chromium and GeoMX, and summarized advantages and disadvantages, focusing on experimental challenges, bioinformatics approaches and the way to interpret cancer biology. The authors analyzed lung cancer, breast cancer and B-cell lymphoma specimens. They described features of each platform and introduce suitable computational approaches for each dataset. However, it is not clear how to make platform decisions. The biological findings which were identified in this study was also not sufficiently described.

Thank you for your feedback. We have significantly revised our manuscript, adding substantial new data comparing the two platforms. This has enabled us to strengthen our discussion on platform selection, providing clearer guidance for decision-making. Additionally, we have enhanced the biological findings by including more detailed analyses to better illustrate the insights gained from our study.

Table 1 is useful for readers but lengthy. It is still not clear how we should choose platforms.

We have refined the table and enhanced the messaging by integrating key insights from the newly generated comparative data. Additionally, we introduced a color-coding system to indicate “positive,” “neutral,” and “negative” categories, allowing for easier assessment across platforms. We aim to convey the suitability of Visium and Chromium for high-throughput, unbiased research, while highlighting the strength of GeoMx for addressing specific questions, albeit with more complex use but flexible design. See page 19 lines 607-625.

In Fig. 1b, DV200 of breast mucinous carcinoma is quite low. Did the RNA quality affect the data quality? The authors should add the discussion about the relationship between sample and data quality to Fig. 1b. Sample ID can be also added to Fig. 1b.

The sample quality is indeed an important factor to consider for transcriptomic assays. All the transcriptome technologies discussed in the manuscript rely on short probes, which are less sensitive to RNA degradation compared to traditional 5' or 3' capture methods. We deliberately chose samples with varying degrees of DV200 and block age to evaluate these technologies in real-life scenarios, particularly for profiling studies that rely on biobank samples. We recognize that this is a crucial question in the field and thank the reviewer for raising it. In response, we have added the sample IDs to Figure 1b as suggested and included additional Supplementary Figures 1b-e comparing gene detection rates with sample age and DV200 values. To our surprise, neither DV200 values nor block age were the main factors affecting quality, as measured by the number of detected genes or gene expression. We have added a comment on this in the manuscript, see page 4, lines 137-139. There was, however, a technical effect observed in one slide of GeoMx (samples L4 and L3), which showed poorer quality which was addressed by batch correction. We added this observation into the manuscript in the methods section together with the batch correction point as in the following chapter. Aside from this, we did not identify other significant technical effects. This suggests there is considerable block-to-block variability, potentially reflecting differences in block collection and storage conditions, which we were unable to assess in this study.

There is no sufficient description and discussion for Fig. 1c. The reviewer wondered why high variation in the number of detected genes were observed (technical batch effects, biological differences, etc. in addition to platform differences).

This observation is greatly appreciated. Biobanked FFPE material can exhibit varying quality due to inherent differences in processing factors such as storage time before fixation, type of fixative, fixation time, and storage conditions (e.g., temperature, humidity). Consequently, we expect data quality to vary between samples. As the reviewer noted, we did observe batch effects for a GeoMx slide (%CH_L_p003/4_WTA) containing samples L3_3 and L4_3. This batch effect was evident in the high-dimensional representation on the TSNE plot (Figure panel g, h). To address this, we applied a batch correction strategy, which effectively mitigated the slide-related effect (Figure panel k) while preserving the biological signal (Figure panel l). An explanation of this approach has been added to the Methods section of the manuscript (page 24, lines 793–803) and as Supplementary Figure 12.

Why was the t-SNE plot used for GeoMX DSP visualization in Fig. 1d? The authors used UMAP plots for Visium and Chromium.

The UMAP plot below shows a similar clustering pattern to the t-SNE plot, primarily driven by batch effects (slide) rather than biological factors of interest (AOI label) prior to batch correction with RUV4. However, the data points appear more cluttered in the UMAP space, making visualization more challenging. We have added the following sentence in the manuscript (see page 24, lines 795-797) to justify our choice: “For the GeoMx data, we used

t-SNE embedding for visualization due to the small number of data points, as it provided a clearer and less cluttered representation. UMAP plots yielded similar conclusions (data not shown).”

GeoMx DSP

Breast

Normalized

Normalized & Batch corrected

Lung

Normalized

Normalized & Batch corrected

DLBCL

Normalized

Normalized & Batch corrected

The authors should indicate numerically or statistically the extent of overlap in comparison between pathology label and deconvolution in Fig. 4a and b.

We agree that this should be addressed more thoroughly. To this end, we have updated Figure 4 by adding panels e and f, which illustrate the consistency between pathologist annotations and deconvolution results on Visium, as well as segment types and deconvolution results on GeoMx. Percentages of agreement scaled per pathology annotation in Visium and AOI type in GeoMx have been added. Additionally, Supplementary Figure 8 has been included to show overlaps between pathology annotations and cell type predictions from deconvolution, with higher resolution and highlights of consensus findings. We hope these additions improve clarity and provide better visualization.

In Fig. 5 and 6, the authors showed several potential therapeutic targets and drugs. The reviewer do not know whether these targets and drugs have been validated and approved. The detailed description and discussion should be needed.

Thank you for this insightful comment. The objective here was to emphasize that a significant number of the identified genes are already targets of existing drugs. To enhance clarity, we have now included detailed information about these genes and their associated drugs in the newly added Supplementary Table 8.

Additionally, we have conducted a more comprehensive analysis comparing the three technologies in assessing these gene targets by computing donor- and technology-specific log-fold change scores (malignant vs. non-malignant cells/spots/ROIs) in newly added Supplementary Table 9. For visualisation of the statistics we added Supplementary Figure 14, showing that the majority of these critical genes for DLBCL are highly differentially expressed between malignant and non-malignant cells/spots and rank among the top 100 genes in at least one donor. We observe that GeoMx exhibits significantly reduced statistical power, largely due to the limited number of data points (ROIs).

It is also worth noting that we observed substantial donor-to-donor variability, further underscoring the utility of these platforms for identifying targeted therapies tailored to individual donors. We have added this information to the text (please see page 15, lines 456-458 and page 19, 584-588), and visualised in newly added Supplementary Figures 10-12.

The authors successfully captured the intra-patient heterogeneity using Chromium or Visium. However, they did not sufficiently mention how we can evaluate this information and use them to identify targeted therapies.

Thank you for addressing this point. We have now added novel results for intra-patient heterogeneity discovery on Figure 5 with GeoMx. The results from all three methods are coherent, finding the two drug targets for both populations between the two areas. The drug target search took place manually by looking up the genes in the Differential Expression list, and comparing this against known drug targets in the ChEMBL database. We have added

this information more specifically into the Methods section (pages 22-23, lines 707-719), as well as added information about these targets in the new Supplementary Table 8.

The reviewer recommend that the authors would include the discussion about the previous studies (e.g. Janesick et al. 2023 Nat Commun; PMID: 38114474) in the context of the comparative analysis of different platforms.

We have expanded the background section in the introduction to better contextualize our work within the existing literature, including the paper referenced by the reviewer. See page 3, lines 86-111.

The reviewer has just visited the github page and confirmed there are enough contents. The reviewer did not install and run the code because the reviewer had no time to do.

We appreciate the time the reviewer took to visit the github page. Thank you. We have now added more code to reflect the previous and current updated analysis.

Reviewer #3

This study conducts a technical comparison of the validity of Visium CytAssist, Chromium Flex, and GeoMx DSP on FFPE samples of different tumors. While such technical comparison is of broad interest, the conclusions are expected. In other words, the current manuscript brings new data to the scientific community but does not bring new insights or knowledge.

We appreciate the reviewer's recognition of the broad interest in our work and respectfully disagree with the suggestion that the conclusions are expected. While GeoMx DSP and Visium are among the most commonly used spatial transcriptomics technologies today, direct head-to-head comparisons remain scarce. To our knowledge, the only existing work is the following unpublished bioRxiv preprint:

Wang et al., "An Experimental Comparison of the Digital Spatial Profiling and Visium Spatial Transcriptomics Technologies for Cancer Research." bioRxiv, April 6, 2023.
<https://doi.org/10.1101/2023.04.06.535805>.

However, this study has significant limitations. It includes only four breast cancer samples and does not perform a direct comparison between Visium and GeoMx on registered adjacent tissue sections, as we do. Additionally, it focuses on a non-specific set of immune AOIs (CD45 and CD8+) and does not integrate snRNA-seq from adjacent slices with rigorous statistical methods.

In contrast, our study profiles on average 24 AOIs per sample covering four types (malignant, T cells, macrophages, and other) from 16 samples spanning three indications (breast cancer, lung cancer, and DLBCL). We captured the range of 822 spots in a Breast

cancer sample to 4951 in DLBCL depending on the size of the tissue with Visium. We also profiled from 802 nuclei to 17,804 nuclei per sample from matched blocks and conducted comprehensive AOI and spot deconvolution and comparative analyses on matched tissue sections using snRNA-seq, Visium, and GeoMx. By leveraging precise mapping of the regions between Visium and GeoMx by image registration, we systematically matched AOIs to individual Visium spots, allowing us to rigorously evaluate the strengths and limitations of each platform.

Among other findings, we identify critical limitations of GeoMx, such as significant background signal from non-selected cell types within pre-selected AOIs and the risk of missing key regions due to the subjective ROI selection process.

Given that Visium and GeoMx remain the technologies of choice for large-scale studies like MOSAIC due to their cost efficiency and robustness, we believe our work fills an important knowledge gap. Our study provides robust, comparative data that will help inform researchers about the practical advantages and limitations of each platform, offering valuable insights to the field.

To further reinforce our findings and message, we have added new data (e.g., Figures 4-6, Supplementary Figures 8-12, 14, Table 1, Supplementary Tables 8 and 9) and significantly improved the manuscript, including updates to the abstract, introduction, and conclusion sections, to better highlight the novelty and impact of our work.

Reviewer #4

Thank you for your time and effort.

Response to the reviewers

We sincerely appreciate the reviewers' thoughtful feedback, which has provided valuable insights to improve our work. After carefully reviewing their comments, we are confident that we have comprehensively addressed the points raised. Specifically:

1. We have summarized molecular events that are identified in each platform based on findings in our data in Table 1 (Reviewer 2).
2. We have added recommendations on combination therapy in the main text (Reviewer 2).

Below, you will find our point-by-point response, where the reviewer's comments are presented in *italic*, followed by our responses in **blue**. We appreciate the opportunity to revise our manuscript and thank you for guiding us through this process. All changes have been highlighted in **yellow** in the revised manuscript.

Reviewer #1

In the revised manuscript, the authors have adequately addressed my comments. Specifically, the results related to Figures 4 and 5 have been updated to strengthen the comparison between the Visium and GeoMX methods. These revisions enhance the novelty of the study and provide insights into platform characteristics, aiding in informed decision-making. (Remarks on code availability)

Thank you for your time and effort.

Reviewer #2

The authors made the effort to add information and descriptions which the reviewer pointed out to the revised manuscript. While the Discussion section was improved, there are still lacks of indication of novelty. For example, in Table 1, if the authors focused on technical comparison in this paper, they should clearly and simply indicate platform differences which were identified by their own study. The current Table 1 seems to be Supplementary information because there are less novel information. The authors should provide a detailed summary of the molecular/cellular events that can only be identified using each platform.

We have added a new category in Table 1 (Disease Biology) to highlight the findings of our paper, and summarize the molecular/cellular events that can only be identified using each platform.

The reviewer could not also clearly understand how we can use the information on intra-tumor heterogeneity to identify targeted therapies. Heterogeneity is an important information on single-cell/spatial transcriptome data. If DE genes in specific clusters were used as target genes, tumor cells would be expected to be only partially eliminated. Should we use common genes among clusters (patient-specific expression) as target genes? (Remarks on code availability)

The list of differentially expressed (DE) genes could help identify targets for therapies, including combinations of existing drugs, potentially reducing the risk of tumor escape.

We have added text in the discussion section to emphasize this point, see lines 612-630.

Reviewer #3

The authors have improved their manuscript extensively and brought new findings in addition to comparisons. It is acceptable now in my opinion. (Remarks on code availability)

Thank you for your time and effort.

Reviewer #4

I co-reviewed this manuscript with one of the reviewers who provided the listed reports. This is part of the Nature Communications initiative to facilitate training in peer review and to provide appropriate recognition for Early Career Researchers who co-review manuscripts. (Remarks on code availability)

The code is organized with a README file providing instructions for installation and reproducibility.

Thank you for your time and effort.